# Chromosome architecture in an archaeal species naturally lacking structural maintenance of chromosomes proteins

Catherine Badel [1,3] ✉ & Stephen D. Bell [1,2] ✉

Proteins in the structural maintenance of chromosomes (SMC) superfamily play key roles in chromosome organization and are ubiquitous across all domains of life. However, SMC proteins are notably absent in the Desulfurococcales of phylum Crenarchaeota. Intrigued by this observation, we performed chromosome conformation capture experiments in the model Desulfurococcales species *Aeropyrum pernix*. As in other archaea, we observe chromosomal interaction domains across the chromosome. The boundaries between chromosomal interaction domains show a dependence on transcription and translation for their definition. Importantly, however, we reveal an additional higher-order, bipartite organization of the chromosome—with a small high-gene-expression and self-interacting domain that is defined by transcriptional activity and loop structures. Viewing these data in the context of the distribution of SMC superfamily proteins in the Crenarchaeota, we suggest that the organization of the *Aeropyrum* genome represents an evolutionary antecedent of the compartmentalized architecture observed in the *Sulfolobus* lineage.

Chromosome architecture has been studied in all three domains of life[1–3]. A common finding across all studied organisms is that proteins belonging to the structural maintenance of chromosomes (SMC) superfamily play pivotal roles in sculpting chromosome conformation[4]. In particular, condensin is near universal with orthologues in all three domains of life[5]. Intriguingly, however, condensin is absent from the Crenarchaeota phylum of Archaea[6]. Our previous work with members of the crenarchaeal Sulfolobales has revealed that a lineage-specific SMC superfamily protein, termed coalescin (ClsN), plays a key role in structuring the chromosomes of these organisms[7,8]. *Sulfolobus* chromosomes have a compartmentalized architecture with A and B domains marked by high and low gene expression, respectively. Elevated ClsN occupancy is causally linked to B compartment identity. In addition to compartmentalization, *Sulfolobus* chromosomes also possess smaller self-interacting domains similar in scale and behaviour to bacterial chromosomal interaction domains (CIDs). Given these similarities,

we have adopted the CID nomenclature for these features of archaeal genomes. In *Sulfolobus*, CIDs are found in both A and B compartments and CID–CID boundaries are principally defined by locally high transcription levels in both compartments. ClsN occupancy is elevated within CIDs in the B compartment. CIDs are also observed in members of the Euryarchaea[9]. In addition, studies in *Haloferax* revealed that deletion of the gene encoding the SMC subunit of condensin had complex effects on chromosome architecture, including a reduction in DNA loops and a loss of CID boundaries across the genome[9]. Thus, in Archaea, as in Bacteria and Eukarya, SMC superfamily proteins play key roles in effecting chromosome architecture. It is therefore of considerable interest that organisms in the Desulfurococcales of phylum Crenarchaeota lack genes encoding SMC superfamily proteins (Extended Data Fig. 1), with the sole exception of the DNA-repair protein, RAD50. Spurred by this observation, we have investigated the chromosome architecture of the Desulfurococcales species *Aeropyrum pernix* K1.

[1]Molecular and Cellular Biochemistry Department, Indiana University, Bloomington, IN, USA. [2]Biology Department, Indiana University, Bloomington, IN, USA. [3]Present address: Génétique Moléculaire, Génomique, Microbiologie, UMR 7156 CNRS, Université de Strasbourg, Strasbourg, France. ✉e-mail: catherine.m.badel@gmail.com; stedbell@iu.edu

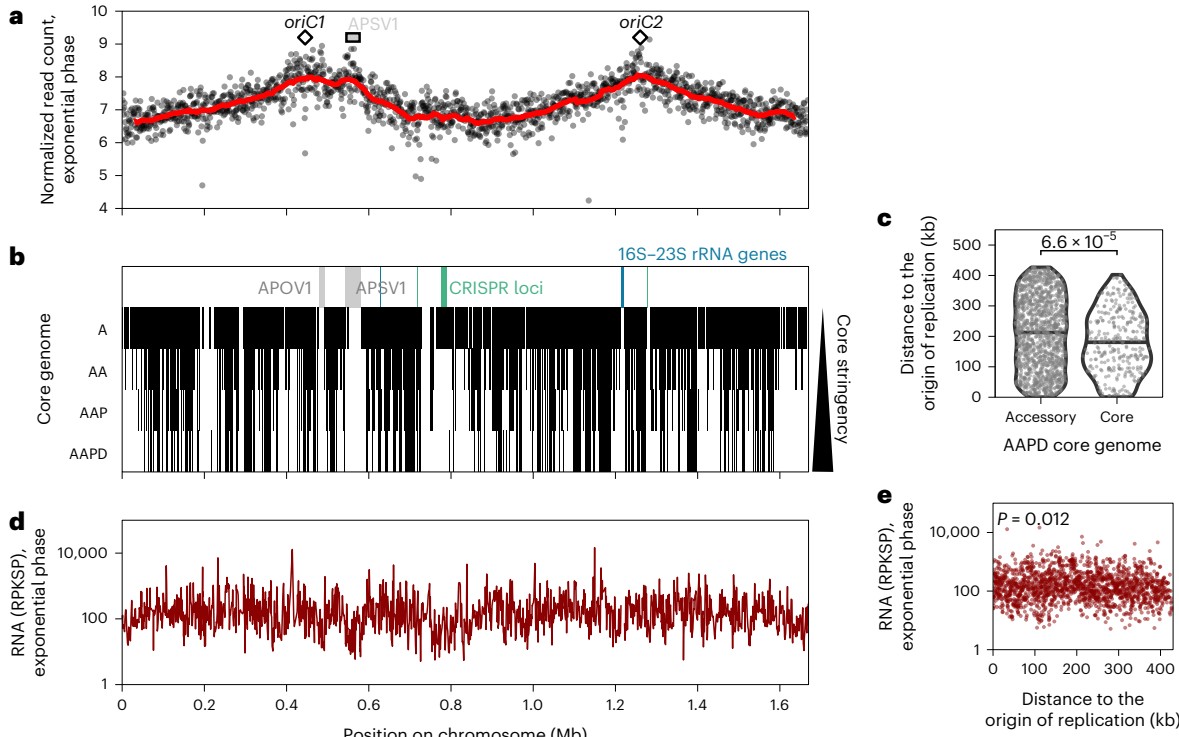

**Fig. 1 | A. pernix primary chromosome organization. a**, MFA in the exponential phase; the red line is a moving point average. **b**, Core gene localization along the chromosome, for core genomes determined with different datasets resulting in different stringency levels (see Supplementary Table 1 for a dataset description). Proviruses are also indicated in grey[41], rRNA genes in blue and CRISPR loci in green. **c**, The distance to the nearest origin of replication of core and accessory genes, at the most stringent level (AAPD). A two-sided Wilcoxon test $P$ value is indicated. **d**, Gene transcriptional level, expressed as RPKSP, in the exponential phase. **e**, Gene transcriptional level plotted in function of the distance to the nearest origin of replication, for the exponential phase. A two-sided Pearson correlation $P$ value is indicated.

## Results

### Primary organization of the A. pernix K1 genome

The 1.669 Mbp genome of *A. pernix* K1 is a closed circle, and a previous candidate locus approach identified two replication origins in the *A. pernix* chromosome[10]. We performed marker frequency analyses (MFA) and confirmed that these two origins are active and that no additional origins exist in this species (Fig. 1a and Supplementary Fig. 1). We note that MFA performed in the stationary phase reveal a marker distribution similar to that of exponentially growing cells. The amplitude of the peaks corresponding to replication initiation is actually greater than that in exponentially growing cells. This striking observation is in agreement with previously published flow cytometry data that revealed an elevated G1- and early-S-phase population in stationary-phase *A. pernix* cells[11]. Examination of gene conservation reveals a significant ($P < 6.6 \times 10^{-5}$) enrichment of core genes in the vicinity of the origins, compared with accessory genes (Fig. 1b,c). We profiled transcription across the chromosome in exponentially growing and stationary-phase cells using RNA sequencing (RNA-seq; Fig. 1d,e, Supplementary Fig. 2 and Extended Data Fig. 2). In exponentially growing cells, there was no significant correlation between distance to origins and transcriptional strength (Fig. 1d,e). In stationary-phase cells, there is a modest but significant ($P = 1.6 \times 10^{-11}$) gradient of enrichment of more highly expressed genes near the origins (Extended Data Fig. 2).

### Chromosome architecture of A. pernix

We performed chromosome conformation capture (3C) experiments on three biological replicates of exponentially growing *A. pernix*. The resulting contact maps, binned at 3 kb resolution, revealed 19 interaction domains that appear, in scale and number, to be analogous to CIDs, along the primary diagonal (to be discussed in more detail below),

in addition to some longer-range interactions (Fig. 2a–e, Supplementary Fig. 3 and Extended Data Fig. 3). Despite analyses of transcript level and gene ontology, no individual distinctive feature could be identified at CID borders. Longer-range interactions were locally depleted, generating some striping in the contact matrix, at 13 loci around the chromosome (for example, at 0.41 Mbp) and at a broader region between ~0.9 Mbp and 1.2 Mbp. The Pearson correlation heat map, and principal component analysis, emphasizes these regions of depleted longer-range interactions (Fig. 2f,g). A previous study in *Haloferax* implicated local regions of AT-rich DNA in the generation of plaid-like patterns on 3C contact maps[9]. However, we could not detect any correlation between nucleotide composition and stripe anchor localization (Extended Data Fig. 4). Analysis of our RNA-seq data reveals that these regions depleted of long-range interaction possess significantly elevated transcription profiles ($P < 2.22 \times 10^{-16}$) compared with the rest of the chromosome (Fig. 2h,i). Accordingly, we will refer to these regions as 'high-expression, insulated domains' (HEIDs). The rest of the chromosome will be referred to as ROC. While *A. pernix* does not encode any candidate SMC proteins, it does encode a RAD50 orthologue. We performed chromatin immunoprecipitation followed by sequencing (ChIP–seq) with antisera that we generated against the recombinant protein and observed a significant enrichment ($P < 2.22 \times 10^{-16}$) of RAD50 within the HEID (Extended Data Fig. 5). In agreement with the elevated transcription in the HEID, genome wide, RAD50 showed a strongly positive (cor = 0.565) and highly significant ($P = 2.4 \times 10^{-281}$) correlation with transcriptional strength. This enrichment of RAD50 at transcriptionally active loci may be related to the documented fragility of active genes in which double-strand breaks can be generated by the processing of R loops[12]. RAD50 is also slightly depleted away from the origin of replications ($P = 2 \times 10^{-15}$).

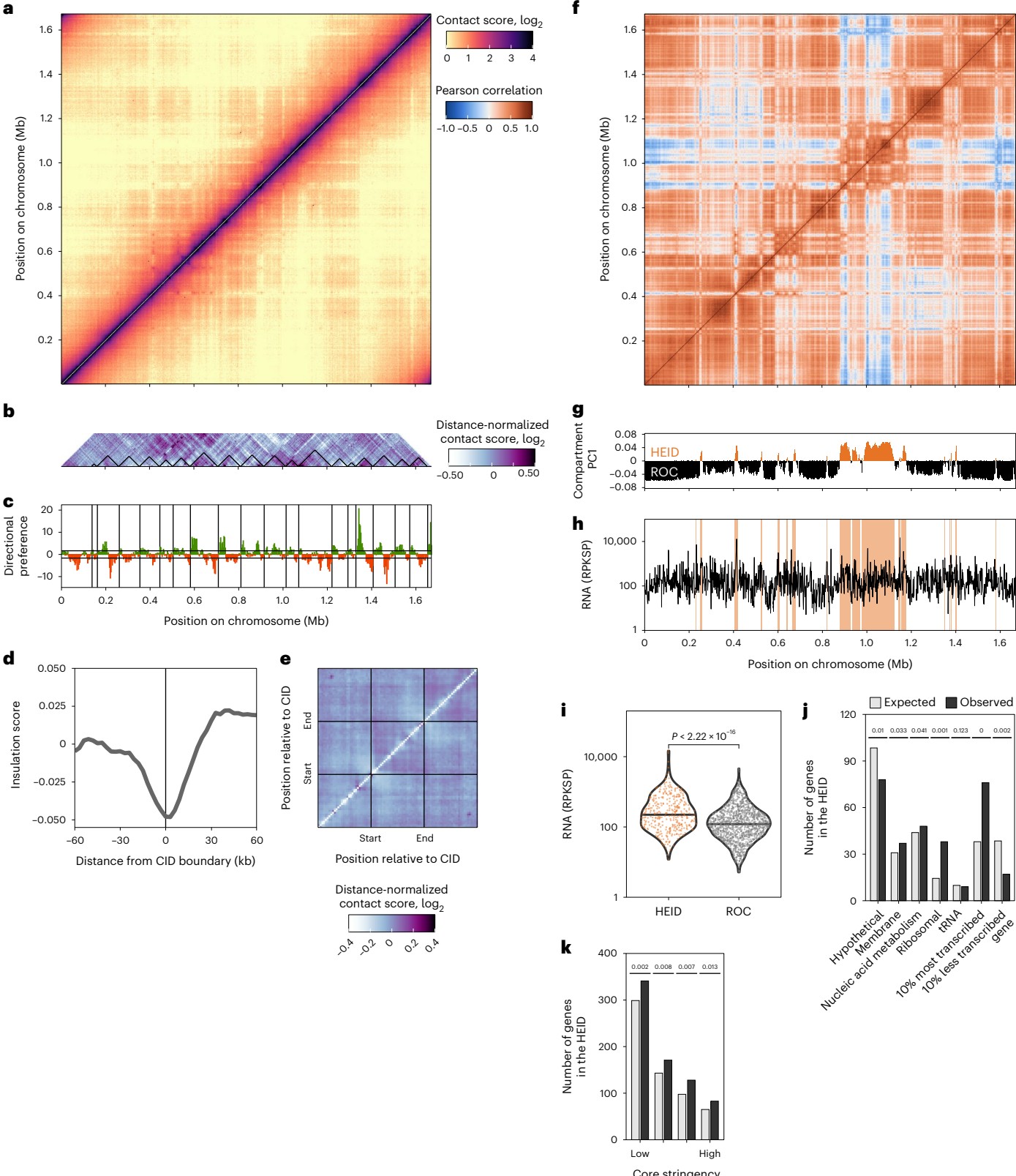

**Fig. 2 | *A. pernix* chromosome is organized into CIDs and a HEID with a higher transcriptional level. a**, Contact score heat map generated at a bin size of 3 kb. **b**, Heat map of the distance-normalized contact score indicating the localization of the CIDs as black triangles. **c**, Directional preference score used to determine the CID boundaries. Positive and negative values of directional preference are in green and orange, respectively. **d**, Aggregate insulation score around CID boundaries. **e**, Aggregate heat map around CIDs. **f**, Pearson correlation heat map at a bin size of 3 kb. **g**, The compartment index (PC1) defines the 'high-expression insulated domain' (HEID) and the ROC; see text for the definition of these features. **h**, The gene transcriptional level (RPKSP) with the HEID highlighted in orange. **i**, Violin plot of the transcriptional level for the HEID and ROC genes. The *P* value of the two-sided Wilcoxon test is indicated, and the horizontal line represents the median. **j**, Number of genes in the HEID, expected from a random distribution of the domains along the chromosome (grey) and observed (black), for different gene groups. An empirical *P* value is indicated (Methods). **k**, The number of core genes in the HEID, expected from a random distribution of the domains along the chromosome (grey) and observed (black), for the various core genomes determined. An empirical *P* value is indicated (Methods).

An analysis of the genes within the HEID reveals an enrichment for those falling within the top 10% of highly transcribed genes, including a significant enrichment ($P = 0.001$) of ribosomal protein genes (Fig. 2j). In addition, genes with low transcript abundance (constituting the 10% lowest expressed) were significantly ($P = 0.002$) depleted from the HEID. The HEID was also significantly enriched in core genes (Fig. 2k).

Next, we used the loop-detection software Chromosight[13], to identify 171 loops in our contact matrices. As can be seen in Fig. 3a, many of the detected loops lie along diagonals with other loops, indicating shared anchor points. Indeed, there is a significant enrichment of loops in clusters of at least six loops (Fig. 3b,c and Extended Data Fig. 6). Furthermore, we observe that loops frequently bridge loci of similar transcriptional levels (Fig. 3e). A total of 38 loops have both anchors within the HEID (Fig. 3d). Conversely, loops are also preferentially anchored at low-transcription loci, potentially suggesting a common silencing mechanism for the two looped loci. In addition, we noted 19 loops emanating from the integrated provirus APSV1, raising the possibility of the provirus using physical proximity to sense the transcriptional status of the cell. Transcription of provirus genes could thus be regulated directly by the formation of hub-like structures—conceivably allowing co-regulation via shared transcription factors or facilitating RNA polymerase recruitment by benefiting from locally high concentrations of the enzyme. Intriguingly, one loop had anchors at APSV1 and a type I-A clustered regularly interspaced short palindromic repeats (CRISPR) array (Fig. 3c) that could indicate the involvement of physical proximity between the CRISPR array and the target DNA for CRISPR adaptation, maturation or interference, or for inhibition by the provirus. Other loops anchored at the CRISPR array could also be involved in the regulation of the CRISPR function. The loop score correlated significantly (cor = 0.398; $P = 2.2 \times 10^{-14}$) with the transcriptional strength of the anchor position (Fig. 3f).

### Transcription, translation and CID strength

In archaea, transcription and translation are believed to be coupled processes[14,15]. Furthermore, treatment of the euryarchaeon *Haloferax volcanii* with the translation inhibitor anisomycin resulted in significant global nucleoid compaction, as visualized using fluorescence microscopy[16]. To test the impact of transcription and translation on chromosome architecture in *Aeropyrum*, we treated cultures with the transcription inhibitor actinomycin D (ActD), in parallel with control cultures treated with the vehicle, DMSO, and with the translation inhibitor chloramphenicol, in parallel with control cultures treated with the vehicle, ethanol (Fig. 4 and Supplementary Fig. 4). ActD led to global transcription inhibition, with an absolute RNA level, measured by reads per kilobase of gene per spike-in (RPKSP), lower than that of the control condition (Fig. 4d, left). However, 91 of 1,753 genes were significantly induced ($P < 0.01$), including proviral hypothetical genes and genes coding for CRISPR proteins, transcription factors, transporters and the chromatin protein Cren7 (log$_2$ fold change (LFC) = 1.47). Upon chloramphenicol treatment, transcription was largely unperturbed with the notable exception of increased transcription of several translation-related genes (Fig. 4d, right). Considered with the growth retardation caused by the chloramphenicol treatment (Supplementary Fig. 4), translation was probably disrupted by the chloramphenicol treatment in *A. pernix*.

Upon transcription inhibition, long-range contacts decreased and short-range contacts increased (Fig. 4a, left, and Fig. 4b). Upon closer inspection of the contact matrix (Fig. 4e, left), we noticed that short-range contacts specifically increased within CIDs. Aggregate contact maps and average insulation scores over CIDs confirmed that CIDs were more insulated in the absence of transcription (Fig. 4e,f, left). Opposite trends were observed upon translation disruption (Fig. 4, right) with long-range contacts slightly increasing and short-range contacts decreasing (Fig. 4a, right, and Fig. 4c). CIDs were less insulated from one another upon chloramphenicol administration (Fig. 4e,f, right, and Fig. 4h). The changes in chromosome conformation were weaker upon chloramphenicol treatment than upon ActD treatment.

### Transcriptional, domain and loop reconfiguration

Upon transcription inhibition, long-range contacts decreased overall but were also reconfigured over the chromosome (Fig. 4a, left). More specifically, changes in long-range contact depletions are evident in the Pearson correlation matrix (Fig. 5a,c). The HEID was disrupted upon ActD treatment, and principal component analyses revealed that a novel HEID′ was formed (Fig. 5b,d). The location of HEID′ correlated with the location of ActD-resistant transcription (Fig. 5d–f), and the RNA level and LFC were significantly higher in the HEID′ than in the ROC (Fig. 5g,h). More specifically, of the 91 genes that are significantly induced upon ActD treatment, 28 (31%) lay in the HEID′, 61 (67%) in the ROC and 2 over both the HEID′ and ROC. This is slightly biased towards the HEID′ compared with the proportion of all genes with 362 of 1,753 (21%) genes in the HEID′ and 1,368 of 1,753 (78%) in the ROC (two-sided Fisher test, $P = 0.0116$).

These data therefore support the hypothesis that gene expression actively structures the HEID. Correlating with the loss of the original HEID on administration of ActD, we saw a loss of HEID-anchored loops (Fig. 5i,j). In addition, we observed the generation of novel loop structures within the new HEID′. We also noted one unanticipated feature of DMSO in the transcriptional induction of a number of loci (Extended Data Fig. 7). Notably, these loci include genes for tetrathionate and polysulfide reductases both of which belong to the DMSO reductase family. Importantly, principal component analyses revealed the DMSO-induced loci to partition with, and thus increase, the HEID of untreated cells and this further correlated with the generation of novel loop structures at these loci. No effect of translation disruption was observed on the HEID and loops (Extended Data Fig. 8).

### Discussion

In the crenarchaeon *A. pernix*, as in other Bacteria and Archaea studied so far[2,3], the chromosome is locally organized in self-interacting CIDs. However, no explanation could be found for the localization of the CID borders, including the frequently observed presence of highly transcribed genes. Our inhibitor studies reveal that CID insulation was decreased by active transcription and increased by translation. Transcription therefore favoured chromosome mixing, while translation impaired it. The opposite role of transcription on the CID admixture was observed in the euryarchaeon *H. volcanii*[9], raising the possibility of diverging mechanisms of CID formation in Crenarchaeota and Euryarchaeota. In the crenarchaeon *A. pernix*, highly expressed loci were aggregated in a HEID and insulated from the ROC. We emphasize that this is occurring in the absence of canonical SMC proteins. RAD50,

---

**Fig. 3 | Chromosomal loops formed between specific loci. a,** Heat map of the distance-normalized contact score at a 3 kb resolution. Loop-type interactions identified by Chromosight[13] are indicated by circles. **b,** Number of loop clusters and number of loops with one or both anchors in a cluster, expected from a random distribution of loops along the chromosome and observed. An empirical $P$ value is indicated (Methods). **c,** Top, the loops are represented as a curve joining the two anchors on a circular chromosome representation, for various loop types. The curve colour represents the loop score. Bottom, aggregate contact maps showing average values of distance-normalized interaction scores around

the loop anchors. **d,** Number of loops, expected from a random distribution of loops along the chromosome (grey) and observed (black), for different gene types found at one or both loop anchors. An empirical $P$ value is indicated (Methods). **e,** Correlation between the local transcriptional level at the two anchor bins of the loops. A two-sided Pearson correlation $P$ value and coefficient are indicated. **f,** Loop score plotted in function of the local transcriptional level at the loop anchor. A two-sided Pearson correlation $P$ value and coefficient are indicated. Cor, correlation coefficient.

the DNA-repair and sole SMC-related protein encoded by *A. pernix*, is probably not involved in the domain formation as it is enriched at transcriptionally active loci throughout the chromosome, not just restricted to HEID loci. Transcription reconfigurations led to changes

in the aggregated loci according to their transcriptional activity, supporting the hypothesis that transcription per se structures the HEID. Punctate contacts, or loops, were enriched in the HEID and were probably involved in the aggregation of the HEID.

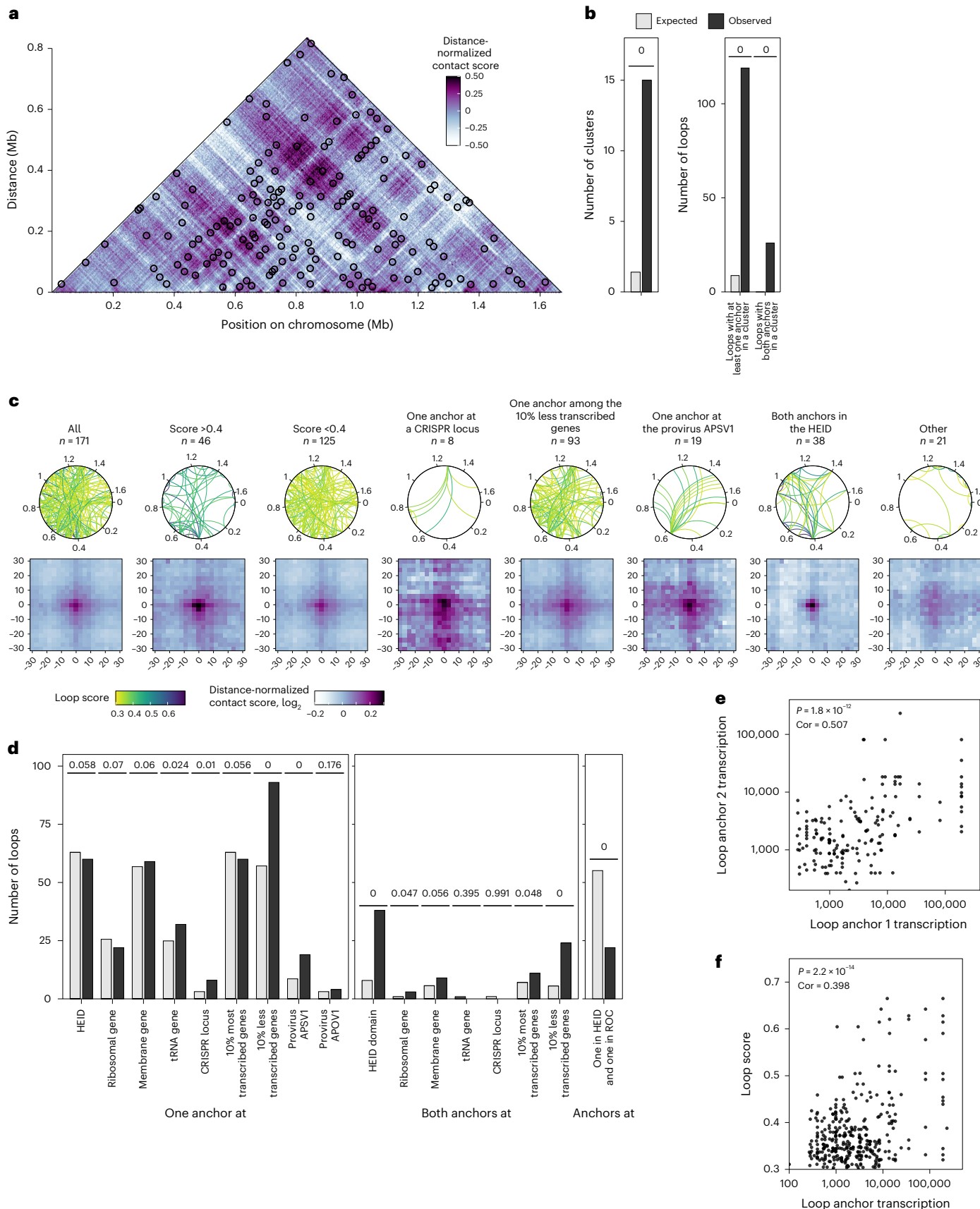

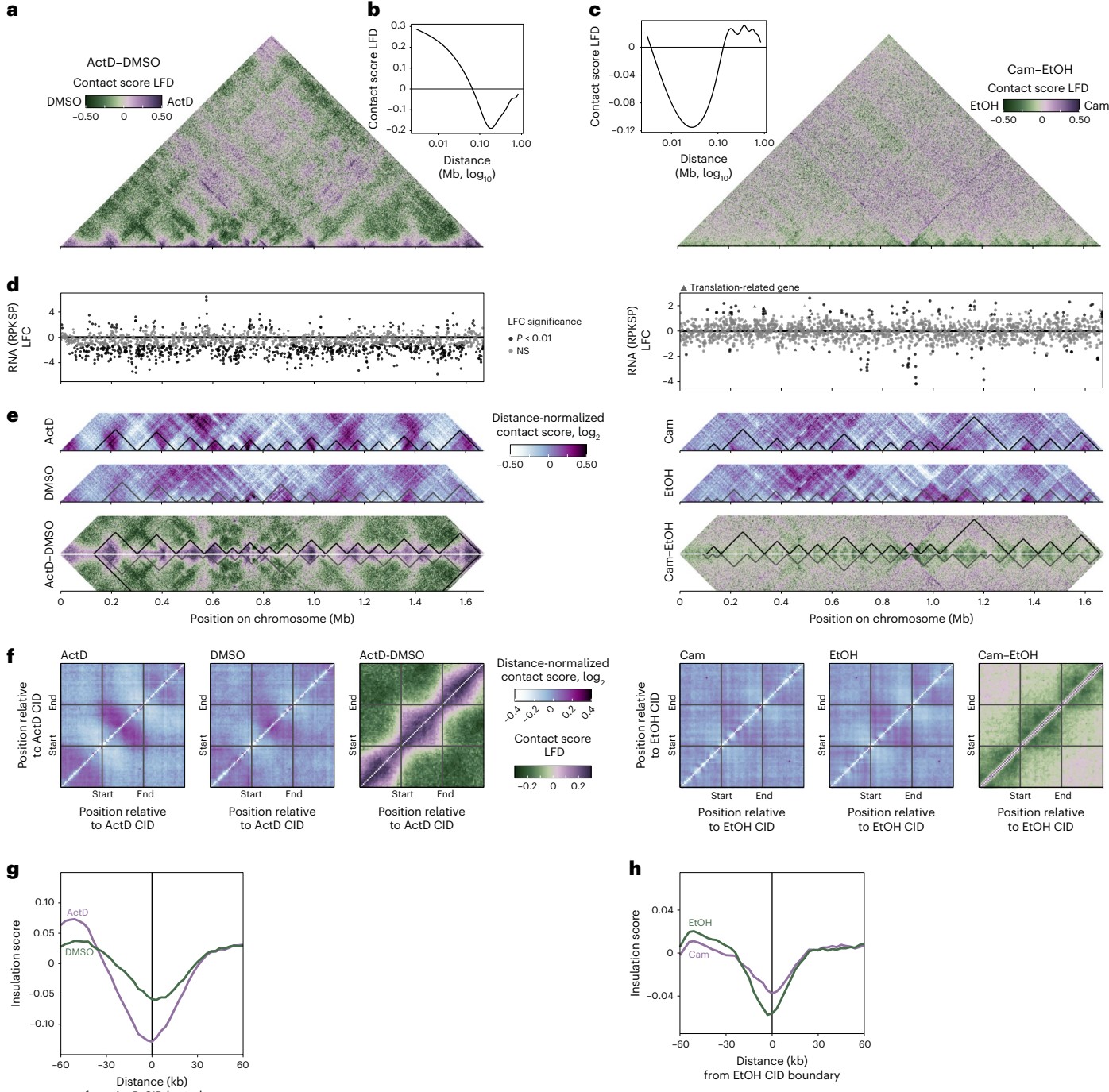

**Fig. 4 | Effect of transcriptional reconfiguration (left, ActD treatment) and translation disruption (right, chloramphenicol treatment) on CIDs.** **a**, Contact score LFD heat maps between the treatment and the control. **b**,**c**, Contact score fold difference between treatment with actinomycin D (**b**) or chloramphenicol (**c**) and control in function of the distance between the interacting bins. Cam, chloramphenicol; EtOH, ethanol. **d**, RNA level (RPKSP) LFC between the treatment and the control. Colours indicate the significance of the change. For the chloramphenicol treatment, triangles indicate translation-related genes. NS, not significant. **e**, Heat maps of the distance-normalized contact score for the treatment and control and heat map of the contact score LFD between the treatment and the control. The heat maps are focusing on contacts between bins that are less than 150 kb apart on the chromosome. The positions of the CIDs are indicated as black triangles for the treatment and grey triangles for the control. **f**, Aggregate heat maps around the CIDs showing the distance-normalized score for the various treatments and controls and the LFD between treatments and controls. **g**,**h**, Aggregate insulation score around the ActD CID boundary (**g**) and the EtOH CID boundary (**h**) for the treatment (purple) and the control (green).

The HEID of the *Aeropyrum* chromosome possesses characteristics reminiscent of the *Sulfolobus* A compartment, including high transcriptional activity and enrichment in ribosomal proteins and conserved genes[17]. Indeed, of the 142 HEID-associated genes in *Aeropyrum* that have clear orthologues in *Sulfolobus acidocaldarius*, 129 are found within the

A compartment in *Sulfolobus* (Fig. 5k). However, contrary to *Sulfolobus*, *A. pernix* does not present a transcriptionally quiescent B compartment, nor does it encode the SMC protein ClsN. These observations, along with the absence of chromosome compartmentalization from all other analysed Archaea[3], lead us to propose the following evolutionary

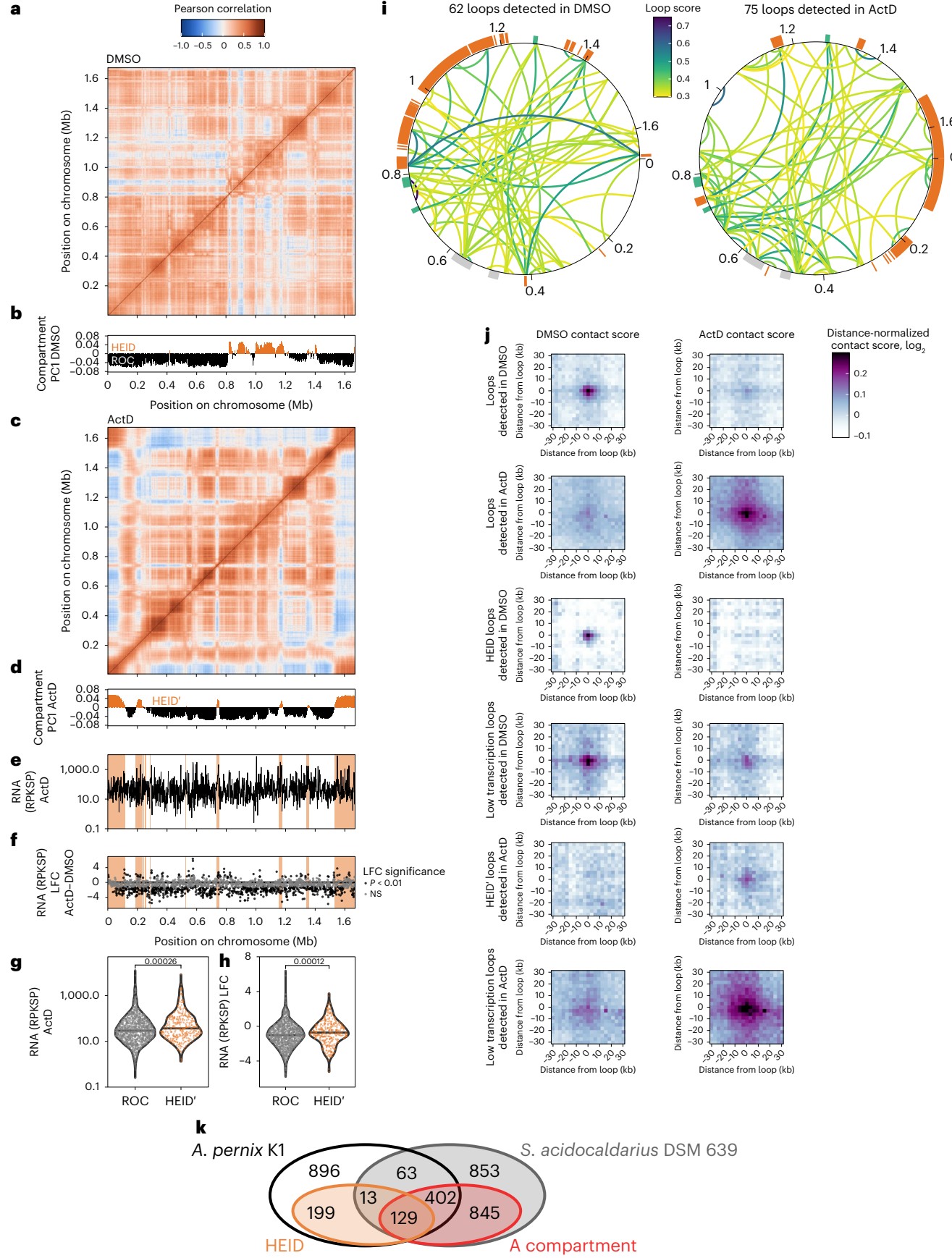

**Fig. 5 | HEID and loop changes upon transcriptional reconfiguration (ActD treatment). a**, Pearson correlation heat map for the DMSO-treated control. **b**, Compartment index (PC1) for the DMSO-treated control. **c**, Pearson correlation heat map for the ActD treatment. **d**, Compartment index (PC1) for the ActD treatment defining a different domain named HEID' (orange). **e**, RNA levels (RPKSP) after ActD treatment. The HEID' is highlighted in orange. **f**, RNA (PRKSP) LFC between the ActD and DMSO treatments. The HEID' is highlighted in orange. **g**, Violin plot of the RNA level for the HEID' and ROC genes. **h**, Violin plot of the RNA LFC for the HEID' and ROC genes. **i**, Loops detected for the DMSO control

and ActD treatments and their score. Various chromosomal structures are indicated as an outside ring: the HEID and HEID' in orange, proviruses in grey and CRISPR loci in green. **j**, Aggregate heat map in DMSO and ActD conditions around the loop anchor, for various categories of loops. **k**, Venn diagram of *A. pernix* K1 and *Sulfolobus acidocaldarius* DSM 639 genes and their chromosomal domain location with the HEID and A compartments highlighted in orange and red for *A. pernix* and *S. acidocaldarius*, respectively. For the violin plots, the *P* value of the two-sided Wilcoxon test is indicated and the horizontal line represents the median.

history of chromosome compartmentalization in the Crenarchaeota. After the loss of the canonical SMC protein Condensin in an ancestor of all present-day Crenarchaeota, constraints on chromosome conformation were lowered, opening the possibility of transcription-mediated aggregation of chromosomal loci in a common ancestor of *Aeropyrum* and *Sulfolobus*. This aggregation leads to the formation of the HEID in *Aeropyrum* and of a HEID-like, proto-A compartment in the *Sulfolobus* ancestor. We suggest that the ancestor of *Sulfolobus* could have then acquired the *clsN* gene by capture of an extrachromosomal element, considering that genes for ClsN-related proteins have been identified in plasmids found in members of the haloarchaea and also in a subset of Asgard archaea[18]. It is notable that the *clsN* gene is encoded within 60 kb of *oriC2* in diverse members of the Sulfolobales such as *Saccharolobus solfataricus* P2, *S. acidocaldarius* DSM 639 and *Sulfolobus islandicus* REY15A. The two replication origins in *A. pernix* correspond to the Orc1-1-dependent *oriC1* and WhiP-dependent *oriC3* of *Sulfolobus* species[19]. Thus, like *clsN*, the Orc1-3-dependent *oriC2* appears to be a Sulfolobales-specific acquisition and is absent from Desulfurococcales. It is possible therefore that *clsN* was acquired along with the *oriC2* replication origin. After the *clsN* gene acquisition at the root of the Sulfolobales, the apparent antagonism between ClsN and transcription[7] would have led to its enrichment in transcriptionally repressed regions of the chromosome and, over evolutionary timescales, have resulted in the formation of the B compartment.

Finally, we return to the central observation that prompted our study—that canonical SMC-based condensin appears to have been lost at the root of the crenarchaeal lineage (Extended Data Fig. 1). In the majority of bacteria, the SMC–ScpAB condensin complex is a key component, along with *parABS* systems, in facilitating the concomitant processes of chromosome replication and segregation[20,21]. This linkage between replication and segregation appears to be found also in the euryarchaeal species that have been investigated[3,22]. Like most bacteria, the euryarchaeal species encode SMC–ScpAB. In contrast, crenarchaea have a fundamentally distinct cell cycle logic, with DNA replication and chromosome segregation temporally separated by gap phases[3]. We therefore hypothesize that the loss of SMC–ScpAB and consequent uncoupling of replication and segregation may have been a key step in the evolution of the distinct cell cycle parameters observed in present-day crenarchaea. Further investigation of cell cycle parameters of diverse archaea will undoubtedly contribute to our understanding of the evolution of cell cycle logics in both archaeal and eukaryotic domains of life.

## Methods

### Strains, media and growth conditions
*A. pernix* K1 (DSM 11879)[23] was obtained from the Leibniz Institute DSMZ-German Collection of Microorganisms and Cell Cultures and grown in homemade Bacto Marine Broth (Difco 2216), supplemented with 1 g l$^{-1}$ of Na$_2$S$_3$O$_3$·5H$_2$O, at 90 °C with agitation. For the transcription inhibition, 60 ml cultures were grown to an optical density at 600 nm (OD600) of 0.3 and treated with 60 µl of 5 mg ml$^{-1}$ ActD diluted in DMSO, 5 µg ml$^{-1}$ final concentration, for 30 min. The control cultures were treated with 60 µl DMSO. For the translation inhibition, 60 ml cultures were grown to OD600 = 0.3 and treated with 441 µl of 34 mg ml$^{-1}$ chloramphenicol diluted in ethanol, final concentration 250 µg ml$^{-1}$,

for 30 min. The control cultures were treated with 441 µl ethanol. For the RNA-seq spike-in normalization, *S. acidocaldarius* DSM 638 was grown in Brock's media[24] containing 0.2% sucrose and 0.1% tryptone, pH 3.2, at 78 °C, with shaking.

### Gene groups in *A. pernix* genome
Several groups of *A. pernix* genes were defined based on the NC_000854.2 annotations and on Gene Ontology terms[25,26]. In details, the hypothetical gene group contained 'hypothetical' in the protein product description. The membrane gene group contained either of the following terms in the protein product description: intramembrane, permease, transporter, secretion system, channel, translocating, translocase, flagellin, flagellar, pilus or pilin. The nucleic acid metabolism group contained DNA or RNA in the protein product description or belonged to the Gene Ontology term 0090304 (nucleic acid metabolic process). A total of 67 ribosomal protein genes were manually selected[27].

### 3C-seq
3C-seq was adapted to *A. pernix* from refs. 8,28. Cells were grown to OD600 -0.3–0.4 and fixed by incubating 40 ml of cell culture in 160 ml of 1× PBS–6% formaldehyde mixture, for 30 min at 25 °C with gentle shaking. The reaction was quenched with 0.5 M glycine (final concentration) for 10 min at room temperature. Fixed cells were collected by centrifugation in protein low-bind tubes. Cells were washed twice with cold 1× PBS and stored at −80 °C. Cells were resuspended and diluted to OD600 = 4 with cold 1× PBS. The suspension (400 µl) was centrifuged, resuspended in 50 µl 1× NEBuffer 2 and treated with 12.5 µl 20% SDS for 15 min at 65 °C, 600 rpm. After a brief cooling on ice, chromosomal DNA was digested by mixing 42 µl of cell lysate with 25.8 µl 10× NEBuffer 2, 120 µl 10% Triton X-100, 97.2 µl H$_2$O and 15 µl of 10 U µl$^{-1}$ AluI (New England Biolabs (NEB)). The reaction was incubated for 3.5 h at 37 °C, 600 rpm, then centrifuged for 20 min at 21,000*g* and 4 °C. The pellet was resuspended in 890 µl H$_2$O and incubated with 100 µl 10× T4 DNA ligase reaction buffer and 10 µl of 400 U µl$^{-1}$ T4 DNA ligase (NEB) at 16 °C, 600 rpm, for 4 h. To reverse cross-links, the ligation reaction was then supplemented with 100 µl 10% SDS, 50 µl 0.5 M EDTA, pH = 8, and 10 µl of 10 mg ml$^{-1}$ proteinase K, and incubated for 6 h at 65 °C and 6 h to 8 h at 37 °C. DNA was extracted twice with phenol:chloroform:isoamyl alcohol and precipitated with isopropanol in the presence of 50 mg glycogen. Purified DNA was resuspended in 40 µl 1× NEBuffer 2 containing 0.1 mg ml$^{-1}$ RNase A and incubated for 30 min at 37 °C. Ligation was confirmed by running 10 µl of purified DNA on an agarose gel. The purified DNA was then extracted again with phenol:chloroform:isoamyl alcohol, precipitated with ethanol and resuspended in 90 µl of 10 mM Tris, pH = 8. DNA was sheared with a Bioruptor (Diagenode) at low power for 40 to 50 cycles (30 s on, 30 s off), and 55.5 µl of the sheared DNA was used to prepare libraries with the NEBNext Ultra DNA Library Prep Kit for Illumina and NEBNext Multiplex Oligos for Illumina (NEB), according to the manufacturer's instructions with size selection for a 300–400 bp insert. DNA libraries were paired-end sequenced on the Illumina NextSeq platform at the Center for Genomics and Bioinformatics at Indiana University.

### 3C-seq contact maps
3C-seq reads were processed using HiC-Pro version 2.9.0 (ref. 29) as performed for *Sulfolobus* species[7]. To adapt HiC-Pro usage to a circular

genome, genomic coordinates were redefined to start at the first AluI restriction site in the genome that is 117 bp from the start of the annotated genome in the public databases. All analyses were performed using this redefined coordinate system. Reads were mapped to this modified genome, and reads resulting from proximity ligation events were counted over 3 kb non-overlapping bins to generate the contact matrix. Intra-bin ligation events were discarded by assigning null values to the matrix diagonal. For the exponential-phase experiment, contact matrix counts were summed over three replicates. For the other experiments, contact matrix counts were summed over two replicates. The data were normalized using the iterative correction and eigenvector decomposition method (ICE correction[30]) with the MAX_ITER parameter of 500. The obtained contact score matrices were further normalized so that the sum of interaction scores was equal to 1,000 for each row and column. Further analysis was performed with R studio build 351, using the tidyverse package, version 1.3.1 (ref. [31]). Distance-normalized Pearson correlation matrices were obtained as described previously[7]. Contact score $\log_2$ fold difference (LFD) matrices were calculated as $\log_2(nijA/nijB)$, where $nij$ is the score of the $i$th row and $j$th column of the matrix for the condition A or B.

For some analyses, to conserve the information about the absolute number of contacts, normalization was performed using DNA abundance along the chromosome in the cell population as described previously[32].

## Aggregate contact maps

Average aggregate contact maps around loop anchors were determined as described previously[8]. For the CIDs, a CID length normalization step to 30 bins (average CID length in *A. pernix*) was included before performing the averaging. In details, for each CID, the adequate matrix was extracted centred on the CID and including the same number of bins as the CID on each side. Linear extrapolation was then used to increase the reduced size of the matrix to 90 bins when necessary. The value was then averaged over all the length-normalized matrices for each position leading to an average matrix that was represented as a contact map.

## Compartment index

The compartment index was calculated as described previously[7], without centreing the Pearson correlation values, using the R package HiTC, version 1.38 (ref. [33]).

## CID analysis

CID boundaries were defined according to the directional preference score as described previously[8] using a distance of 60 kb. The insulation score was also calculated as described previously[7]. CID boundaries were explored visually to look for potential common features including gene orientation and transcriptional level. Different parameters were compared using violin plots and Wilcoxon tests between border bins and non-border bins, including RNA level, relative enrichment of GC basepairs and RAD50 enrichment and evidencing no statistical differences. Permutation tests were also performed on CID localization to test whether loop anchors or certain gene types were enriched at CID boundaries, evidencing no statistical difference. Note that the 3 kb resolution of the analysis might prevent determining a small distinctive feature of the CID boundaries.

## Loop analysis

Loops were identified for each condition using Chromosight[13] as previously described[8], retaining all the loops detected with a score higher than 0.3. Note that the number of reads used in Chromosight influences the power of loop detection, explaining why less loops are detected after the ActD and chloramphenicol treatments (two replicates pooled) compared with the exponential-phase sample (three replicates pooled). Loop clusters were called when there were five or more loops anchored in the same bin or six or more loops anchored in three consecutive bins.

## Statistical analysis

Two-sided Wilcoxon tests were used to compare the distance to the origin of replication, the RNA level and the RAD50 enrichment of genes, and the GC richness of bins from the HEID' and the ROC. A non-parametric test was chosen because the variables are not normally distributed. We assumed the independence and equal variance of the tested parameters between genes.

## Permutation tests

Various randomized permutation tests were performed to test whether specific genomic parameters were not randomly distributed along the chromosome. For all cases, the permutation procedure was repeated 1,000 times, and the expected value was determined as the mean value of all the repeats for the genomic parameter of interest. To compute an empirical *P* value, we divided the number of permutation procedures in which the simulated value was the same as the real observed value by the number of the repeats.

To determine whether genes are randomly distributed in the HEID, we randomly permuted the localization of the HEID segments, keeping the same segment number and length as observed. The number of genes from various gene groups located in the HEID was counted. To determine whether loops were more clustered than randomly expected and whether loops were anchored at specific gene groups, we randomly permuted the localization of the loops, keeping the same loop number and length. A loop was considered anchored at a specific gene group if at least one of the group members was present in the anchor bin.

## MFA−seq

MFA was performed using Illumina-based next-generation sequencing. DNA was extracted from exponentially growing cells and the stationary-phase population, according to a previous study[23] with modifications. The culture (10 m) was centrifuged and resuspended in 300 µl NET buffer (50 mM Tris, pH = 8, 100 mM EDTA, 150 mM NaCl). Cells were lysed by adding 220 µl lysis buffer (50 mM Tris, pH = 8, 100 mM EDTA, 150 mM NaCl and 5% SDS), supplemented with 1 µl of 10 mg ml⁻¹ RNase A and incubated for 20 min at room temperature. The mixture was then supplemented with 5 µl of 10 mg ml⁻¹ proteinase K and incubated for 30 min at 65 °C. DNA was extracted at least twice with phenol:chloroform:isoamyl alcohol and once with chloroform, precipitated with ethanol and resuspended in 200 µl of 10 mM Tris, pH = 8. DNA libraries were prepared with the Nextera XT DNA Library Preparation Kit and paired-end sequenced on the Illumina NextSeq platform at the Center for Genomics and Bioinformatics at Indiana University. Read counts for exponentially growing cells were grouped into 1 kb bins. Normalization was performed as in ref. [34] to account for GC biases in sequencing. For each condition, the GC bias was modelled using a linear regression fitting the data (Supplementary Fig. 1). For each bin $i$, the normalized read count $n_{i,\text{normalized}}$ was calculated as $n_{i,\text{normalized}} = n_{i,\text{observed}} - (n_{i,\text{theoretical}} - n_{\text{average}})$, where $n_{i,\text{observed}}$ is the observed number of reads for the bin $i$, $n_{i,\text{theoretical}}$ is the theoretical read count for the bin $i$ calculated from the linear regression and $n_{\text{average}}$ is the average number of reads across all the bins.

## RNA-seq

RNA extraction was adapted to *A. pernix* from ref. [35], and spike-in normalization was adapted from ref. [36] to compare between samples. *A. pernix* (Ape) culture (10 ml) was mixed with an appropriate volume of stationary-phase *S. acidocaldarius* (Sac) such that volume Sac = 0.01 × volume Ape × OD600 Ape/OD600 Sac. The culture mixture was passed through a 0.45 µM nitrocellulose filter. The filter was then placed in a microcentrifuge tube containing 600 µl lysis buffer (100 mM sodium acetate, pH = 5.2, and 2% SDS) and 600 µl phenol, pH = 4.3. The tube was vortexed for 2 min and centrifuged for 2 min at 14,900*g*. The aqueous phase was then extracted at least one additional time with acid phenol. The nucleic acids were precipitated with

isopropanol, resuspended in 51 µl $H_2O$ and treated with DNase (Invitrogen, amplification grade) according to the manufacturer's instructions. The extracted RNA was further precipitated with isopropanol and resuspended in 20 µl $H_2O$. The purified RNA was then directly used to prepare strand-specific libraries with the NEBNext Ultra II Directional RNA Library Prep Kit for Illumina (NEB) according to the protocol for purified mRNA or ribosomal RNA (rRNA)-depleted RNA in the manufacturer's manual. The libraries were paired-end sequenced on the Illumina NextSeq platform at the Center for Genomics and Bioinformatics at Indiana University.

### RNA-level analysis
Reads were mapped to *A. pernix* (accession NC_000854.2) and *S. acidocaldarius* (NC_007181.1) using Bowtie 2, version 2.4.1, with default parameters[37] and counted using SeqMonk version 1.48, either over non-overlapping 3 kb windows or over genes. The read count was identical whether reads were aligned in parallel or sequentially to both chromosomes, except for the genes coding for 16S and 23S rRNA. Bowtie2 optimization efforts to differentiate between *Aeropyrum* and *Sulfolobus* rRNA genes were unsuccessful. Those genes or the windows containing them were therefore removed from further analysis. For window analysis, the spike-in normalization parameter was the average raw window count of *S. acidocaldarius* and the normalized read count (reads per spike-in) for *A. pernix* was the raw window count divided by the spike-in normalization parameter. For the gene analysis, the spike-in normalization parameter was the read-per-kilobase value, averaged over all genes of *S. acidocaldarius* and the normalized read count RPKSP for *A. pernix* was the read-per-kilobase value divided by the spike-in normalization parameter. Results were then averaged over replicates.

Differential RNA levels were analysed using the R package DESeq2 version 1.34 (ref. 38), using spike-in normalized RPKSP values. The raw LFC and adjusted *P* value were subsequently used.

### ChIP–seq
ChIP–seq was adapted to *A. pernix* from ref. 39. Cells were grown to OD600 ~0.3. The culture (40 ml) was cross-linked with either 1% or 2.5% formaldehyde for 20 min at room temperature. The reaction was quenched with 100 mM and 300 mM glycine, respectively, for 10 min at room temperature. Fixed cells were collected by centrifugation, washed with 20 ml cold 1× PBS and resuspended in TBS-TT (20 mM Tris, pH = 7.4, 150 mM NaCl, 0.1% Tween-20 and 0.1% Triton X-100). Chromatin was fragmented with a Bioruptor (Diagenode) at medium power for 25 cycles (30 s on, 30 s off), and the extract was clarified by centrifugation. Immunoprecipitation was then performed as described in ref. 7. After phenol:chloroform:isoamyl alcohol extraction and isopropanol precipitation, immuno-precipitated DNA was resuspended in 50 µl TE. A total of 50 µl of ChIP reactions and 100 pg of input DNA were used to prepare libraries with the NEBNext Ultra II Library Prep Kit (NEB) according to the manufacturer's instruction. DNA libraries were paired-end sequenced on the Illumina NextSeq platform at the Center for Genomics and Bioinformatics at Indiana University.

Reads were mapped to the *A. pernix* genome using Bowtie 2, version 2.4.1, with default parameters[37] and counted using SeqMonk, version 1.48, for 500 bp non-overlapping windows. ChIP–seq coverage was divided by input coverage after normalizing for the total number of reads mapped to the chromosome. Correlation between the two fixation methods was high, and the two methods were considered as replicates and their score averaged. Analyses were also performed with each individual replicate and yielded similar results.

### Core genome analysis
The core genome was determined at four different taxonomic levels (*Aeropyrum* only (A), *Aeropyrum* and *Acidolobus* (AA), *Aeropyrum*, *Acidolobus* and *Pyrodictiaceae* (AAP), *Aeropyrum*, *Acidolobus*, *Pyrodictiaceae* and the rest of the *Desulfurococcaceae* (AAPD); Supplementary

Table 1) using Get_homologues for protein-coding genes[40]. Clustering was performed with both orthoMCL and COGtriangle with standard parameters. Core clusters were the ones with proteins in all genomes for both methods.

### Venn diagram analysis
Orthologous protein-coding genes shared by *A. pernix* K1 and *S. acidocaldarius* DSM 639 were determined using Get_homologues[40] using the orthoMCL algorithm with standard parameters.

### Reporting summary
Further information on research design is available in the Nature Portfolio Reporting Summary linked to this article.

## Data availability
All sequencing data have been submitted to the NCBI Sequence Read Archive (SRA). Submission ID: SUB13894161. BioProject ID: PRJNA1027590; http://www.ncbi.nlm.nih.gov/bioproject/1027590.

## Code availability
No custom code was generated for this work.

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

## Acknowledgements
This work was funded by NIH grant R01 GM135178 and R35 GM152171 (S.D.B.) and the College of Arts and Sciences, Indiana University (S.D.B.).

## Author contributions
S.D.B. supervised the research and obtained research funding. S.D.B. and C.B. conceived the study and designed the experiments. C.B. performed the experiments, analysed the data and prepared the figures. C.B. wrote the first draft of the paper. S.D.B. and C.B. edited and reviewed the paper.

## Competing interests
The authors declare no competing interests.

## Additional information
**Extended data** is available for this paper at https://doi.org/10.1038/s41564-023-01540-6.

**Correspondence and requests for materials** should be addressed to Catherine Badel or Stephen D. Bell.

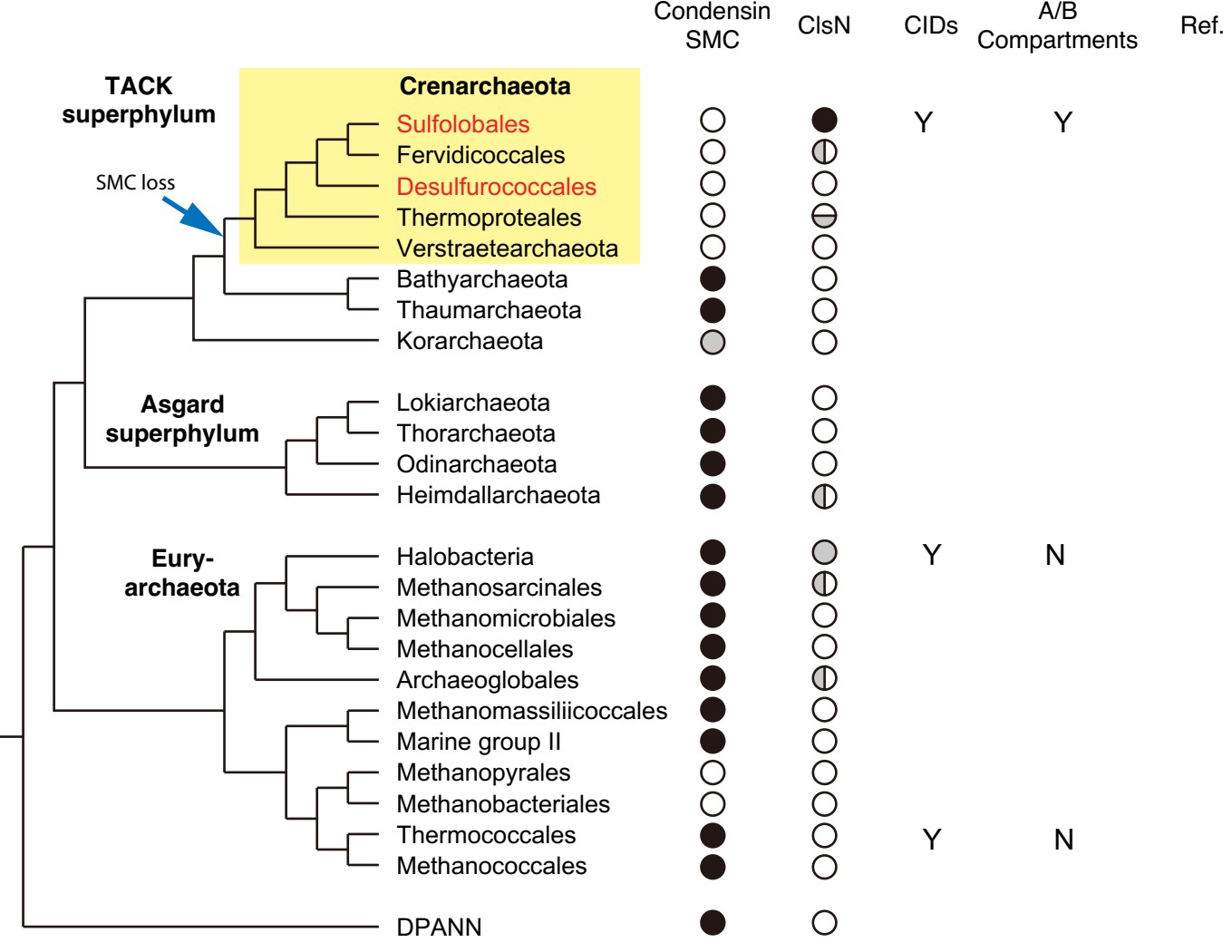

**Extended Data Fig. 1 | Distribution of SMC superfamily proteins in archaeal species.** Adapted from[7]. The positions of the Sulfolobales and Desulfurococcales (of which *Aeropyrum pernix K1* is a member) are indicated in red within the yellow shaded box indicating the crenarchaea. *Aeropyrum* was selected as our species of choice due to the ease of growth in an aerobic environment and the availability of a high-quality, fully-closed genome sequence for the species. For a comprehensive analysis of the distribution of ClsN-like proteins in the archaea the reader is directed to ref. 18.

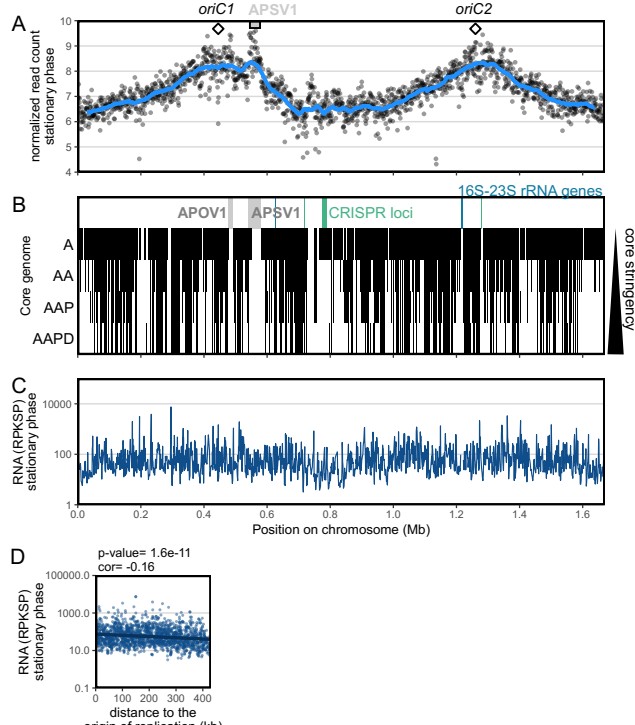

**Extended Data Fig. 2 | Primary chromosome organization in stationary phase *Aeropyrum pernix*. a** Marker Frequency Analysis in stationary phase. **b**. Core gene localization along the chromosome, for core genomes determined with different datasets resulting in different stringency levels **(see Table S2 for dataset description)**. Proviruses are also indicated in grey (Mochizuki et al.[41]), rRNA genes in blue and CRISPR loci in green. **c** Gene transcriptional level, expressed as RPKSP stationary phase. **d**. Gene transcriptional level plotted in function of the distance to the nearest origin of replication, for stationary phase. Two-sided Pearson-correlation p-value and coefficient are indicated.

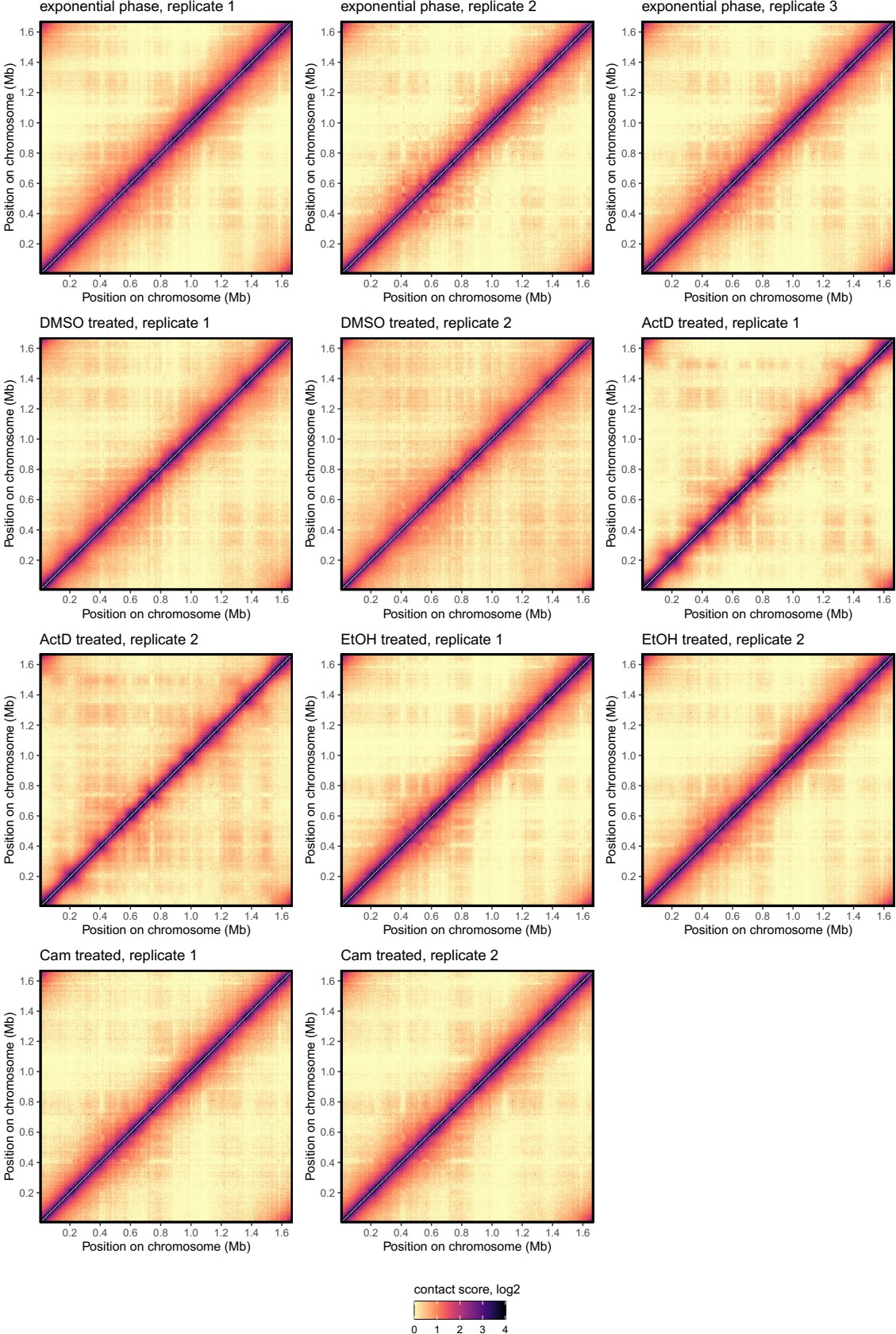

**Extended Data Fig. 3 | Individual contact score heatmaps.** Data generated at a bin size of 3 kb for all the conditions presented in this study.

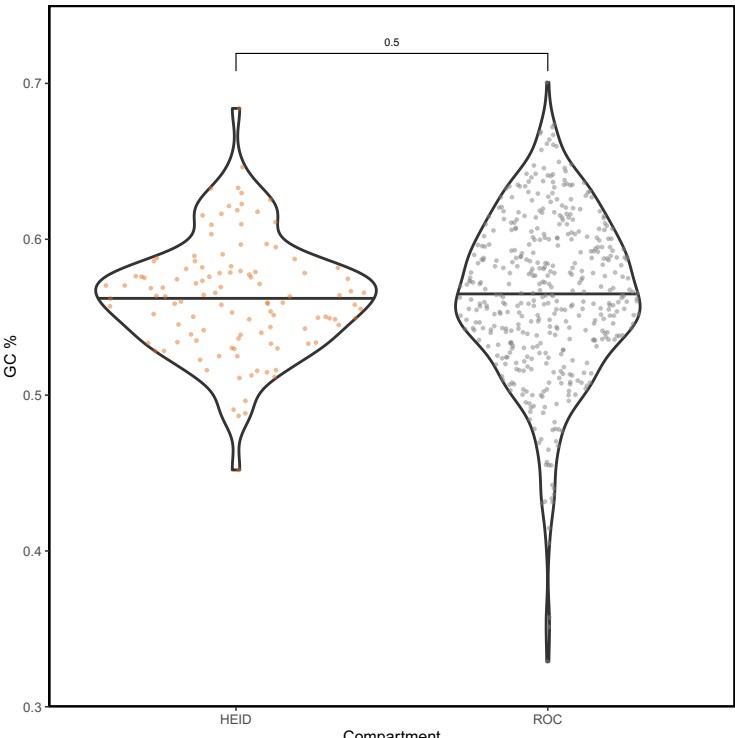

**Extended Data Fig. 4 | Violin plot of the GC content (%) in 3 kb bins between the HEID domain and ROC.** The p-value of two-sided Wilcoxon test is indicated and the horizontal line represents the median.

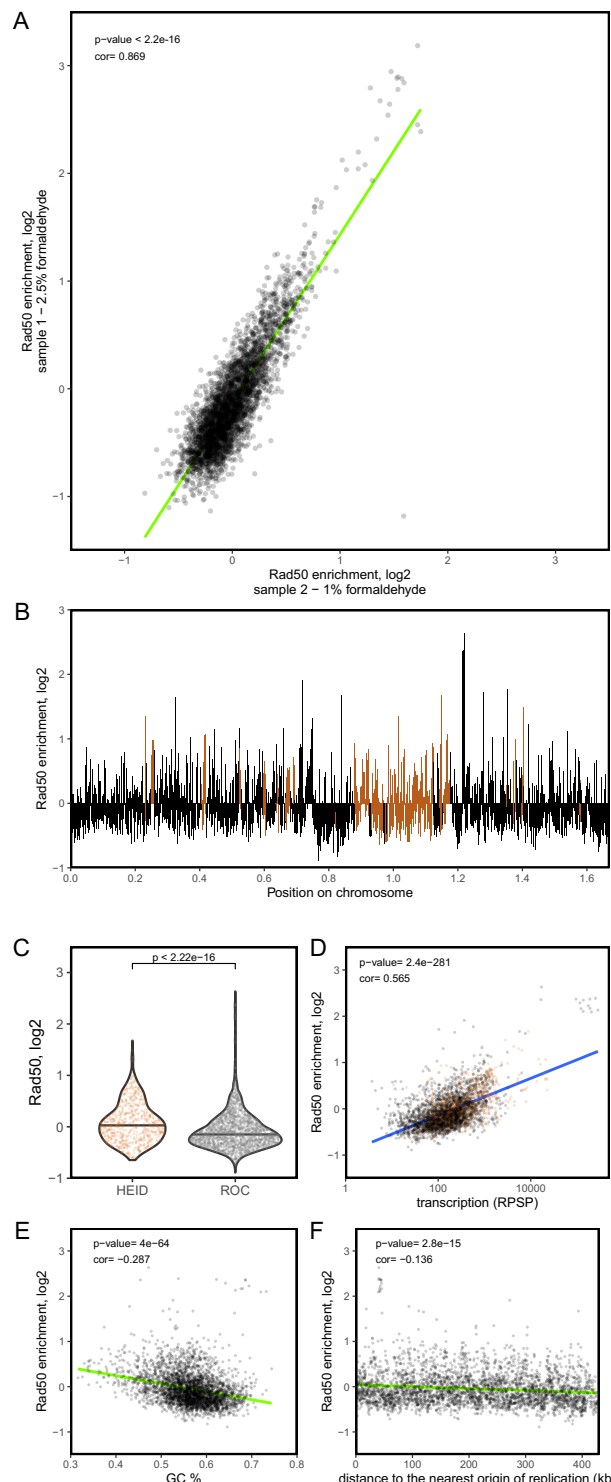

**Extended Data Fig. 5 | ChIPseq analysis of RAD50 binding to Aeropyrum pernix chromosome. a**. Reproducibility between the two experimental conditions used. **b**. RAD50 enrichment per 3 kb window along the chromosome with the HEID domain highlighted in orange. **c**. Violin plot of RAD50 enrichment in the HEID domain and ROC. The p-value of two-sided Wilcoxon test is indicated and the horizontal line represents the median. **d**. Correlation between RAD50 enrichment and the transcriptional level (RPSP) per 3 kb window. **e**. Correlation between RAD50 enrichment and the GC content per 3 kb window. **f**. Correlation between RAD50 enrichment and the distance to the origin of replication. Two-sided Pearson correlation p-value and coefficient are indicated. For A, D, E and F, Two-sided Pearson correlation p-values and estimates were indicated.

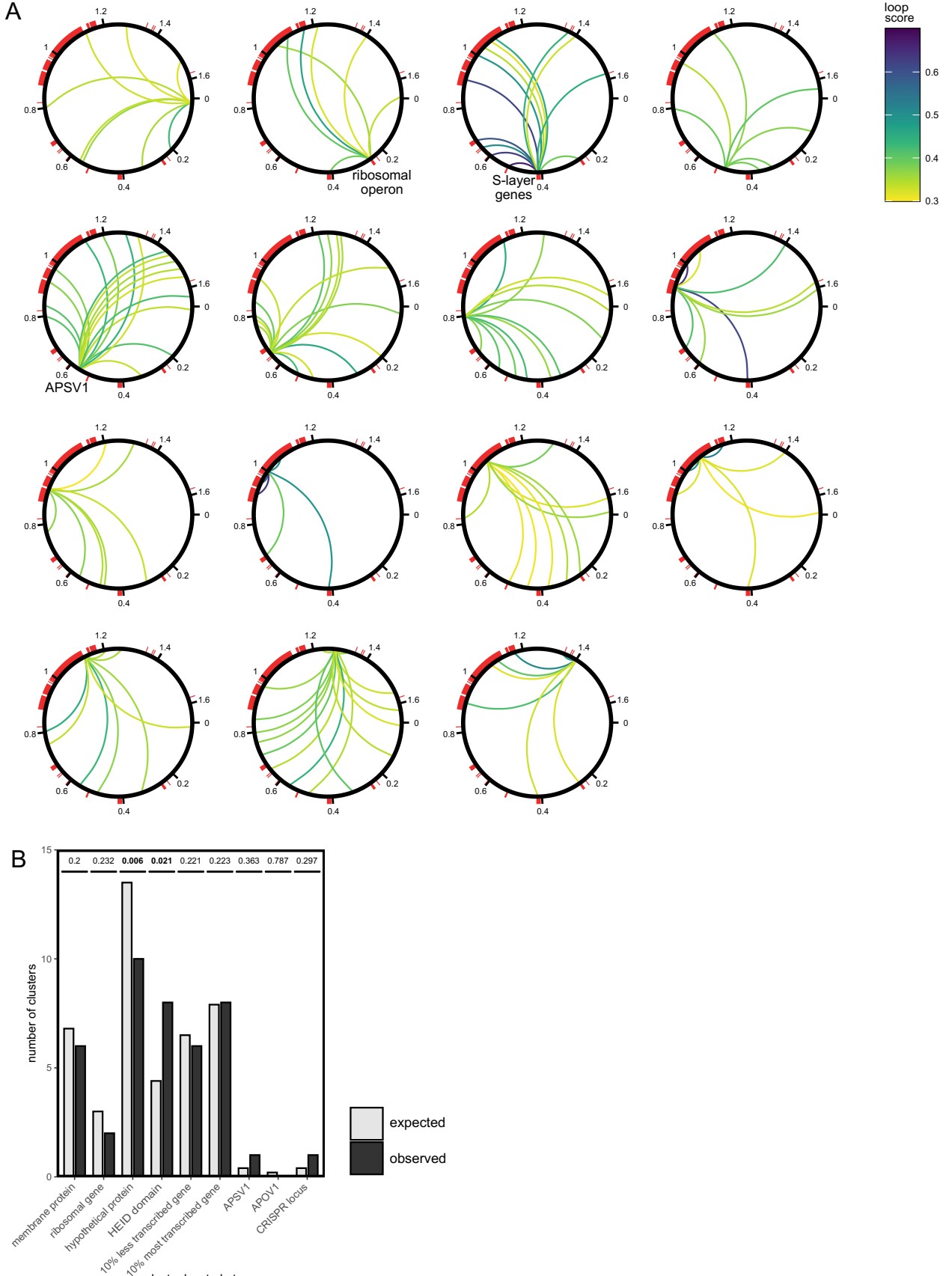

**Extended Data Fig. 6 | The 15 loop clusters.** A The loops of each cluster are represented as a curve joining the two anchors on a circular chromosome representation. The HEID domain in indicated in orange. For three clusters, an interesting genetic element is located at the cluster anchor and is indicated.

B Number of clusters, expected from a random distribution of clusters along the chromosome (grey) and observed (black), for different genetic elements found at the cluster anchor. An empirical p-value is indicated (see material and method).

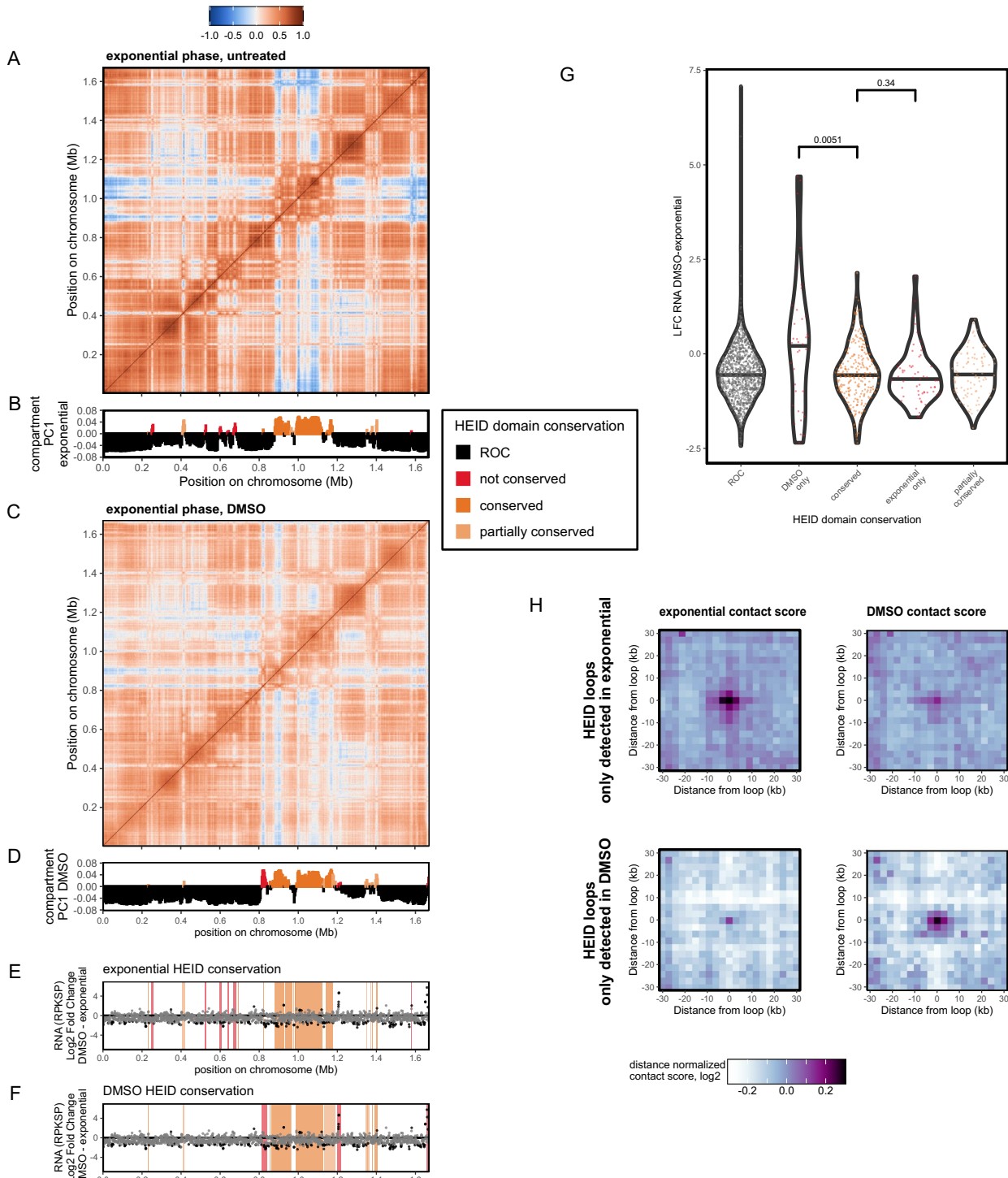

**Extended Data Fig. 7 | HEID domain and loop changes between untreated exponential phase sample and DMSO-treated exponential phase sample.**
**a**. Pearson correlation heatmap for the untreated sample. **b**. Compartment index (PC1) for the untreated sample. The conservation of the HEID domain segments with the DMSO-treated sample is indicated in orange shades.
**c**. Pearson correlation heatmap for the DMSO treated sample. **d**. Compartment index (PC1) for the DMSO treatment. The conservation of the HEID domain segments with the untreated sample is indicated in the same orange shades as in B. **e**. RNA (PRKSP) Log2 Fold Change (LFC) between the DMSO-treated and untreated samples. The conservation of the untreated HEID domain segments

with the DMSO-treated sample is indicated in the same orange shades as in B. **f**. RNA (RPKSP) Log2 Fold Change (LFC) between the DMSO-treated and untreated samples. The conservation of the DMSO-treated HEID domain segments with the untreated sample is indicated in the same orange shades as in B. **g**. Violin plot of the RNA LFC in function of the HEID domain segment conservation between the untreated and DMSO-treated samples. The p-value of two-sided Wilcoxon test is indicated and the horizontal line represents the median. **h**. Aggregate heatmap in untreated and DMSO-treated conditions around the loop anchor, for various categories of loops.

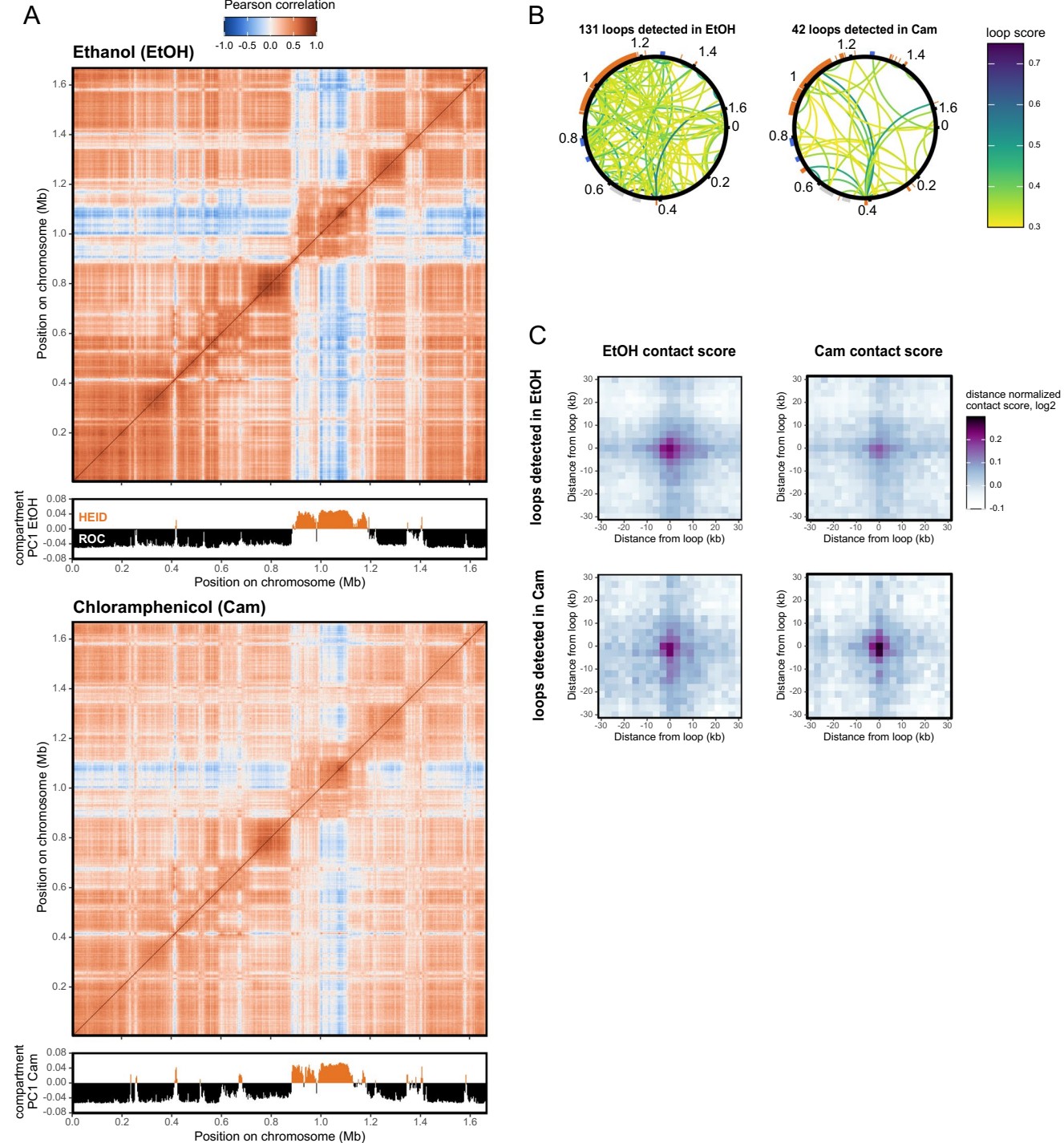

**Extended Data Fig. 8 | HEID domain and loop changes upon translation disruption (Chloramphenicol treatment). a.** From top to bottom. Pearson correlation heatmap for the ethanol (EtOH)-treated control. Compartment index (PC1) for the ethanol-treated control. Pearson correlation heatmap for the Chloramphenicol (Cam) treatment. Compartment index (PC1) for the

Chloramphenicol treatment. **b.** Loops detected for the ethanol control and chloramphenicol treatments and their score. Various chromosomal structures are indicated as an outside ring: the HEID domain in orange, proviruses in grey and CRIPSR loci in green. **c.** Aggregate heatmap in Ethanol and chloramphenicol conditions around the loop anchor, for various categories of loops.

# Reporting Summary

Please do not complete any field with "not applicable" or n/a.  Refer to the help text for what text to use if an item is not relevant to your study.
For final submission: please carefully check your responses for accuracy; you will not be able to make changes later.

## Statistics

For all statistical analyses, confirm that the following items are present in the figure legend, table legend, main text, or Methods section.

| n/a | Confirmed | |
|---|---|---|
| ☐ | ☑ | The exact sample size (*n*) for each experimental group/condition, given as a discrete number and unit of measurement |
| ☒ | ☐ | A statement on whether measurements were taken from distinct samples or whether the same sample was measured repeatedly |
| ☐ | ☑ | The statistical test(s) used AND whether they are one- or two-sided<br>*Only common tests should be described solely by name; describe more complex techniques in the Methods section.* |
| ☐ | ☑ | A description of all covariates tested |
| ☐ | ☑ | A description of any assumptions or corrections, such as tests of normality and adjustment for multiple comparisons |
| ☐ | ☑ | A full description of the statistical parameters including central tendency (e.g. means) or other basic estimates (e.g. regression coefficient) AND variation (e.g. standard deviation) or associated estimates of uncertainty (e.g. confidence intervals) |
| ☐ | ☑ | For null hypothesis testing, the test statistic (e.g. *F*, *t*, *r*) with confidence intervals, effect sizes, degrees of freedom and *P* value noted<br>*Give P values as exact values whenever suitable.* |
| ☒ | ☐ | For Bayesian analysis, information on the choice of priors and Markov chain Monte Carlo settings |
| ☒ | ☐ | For hierarchical and complex designs, identification of the appropriate level for tests and full reporting of outcomes |
| ☐ | ☑ | Estimates of effect sizes (e.g. Cohen's *d*, Pearson's *r*), indicating how they were calculated |

*Our web collection on statistics for biologists contains articles on many of the points above.*

## Software and code

Policy information about availability of computer code

| | |
|---|---|
| Data collection | Sequencing was performed at the Center for Genomics and Bioinformatics at Indiana University, Illumina NextSeq platform. |
| Data analysis | HiC-Pro version 2.9.0; R studio build 351, tidyverse package version 1.3.1, package HiTC version 1.38, package DESeq 2 version 1.3 4; Chromosight 1.3.1; Bowtie 2, version 2.4.1; SeqMonk version 1.48; Get_homologues 3.6.1; FastQC 0.11.3 |

For manuscripts utilizing custom algorithms or software that are central to the research but not yet described in published literature, software must be made available to editors and reviewers. We strongly encourage code deposition in a community repository (e.g. GitHub). See the Nature Portfolio guidelines for submitting code & software for further information.

## Data

Policy information about availability of data

All manuscripts must include a data availability statement. This statement should provide the following information, where applicable:
- Accession codes, unique identifiers, or web links for publicly available datasets
- A description of any restrictions on data availability
- For clinical datasets or third party data, please ensure that the statement adheres to our policy

Data deposited at NIH SRA BioProject ID PRJNA1027590: http://www.ncbi.nlm.nih.gov/bioproject/1027590

# Research involving human participants, their data, or biological material

Policy information about studies with [human participants or human data](). See also policy information about [sex, gender (identity/presentation), and sexual orientation]() and [race, ethnicity and racism]().

| | |
|---|---|
| Reporting on sex and gender | NA |
| Reporting on race, ethnicity, or other socially relevant groupings | NA |
| Population characteristics | NA |
| Recruitment | NA |
| Ethics oversight | NA |

Note that full information on the approval of the study protocol must also be provided in the manuscript.

# Field-specific reporting

Please select the one below that is the best fit for your research. If you are not sure, read the appropriate sections before making your selection.

☑ Life sciences  ☐ Behavioural & social sciences  ☐ Ecological, evolutionary & environmental sciences

For a reference copy of the document with all sections, see [nature.com/documents/nr-reporting-summary-flat.pdf](nature.com/documents/nr-reporting-summary-flat.pdf)

# Life sciences study design

All studies must disclose on these points even when the disclosure is negative.

| | |
|---|---|
| Sample size | Depending on analysis, experiments were performed in duplicates or triplicates. No sample size calculation was performed, duplicate and triplicate are standerd sizes for this sort of analysis. |
| Data exclusions | No data were excluded |
| Replication | Pearson correlation between replicate results was tested and high (see Supplementary Information for examples of replication |
| Randomization | No randomization was performed, not approriate to the data under analysis. |
| Blinding | Data inappropriate for blinding during analyses (largely intra-sample analyses) |

# Behavioural & social sciences study design

All studies must disclose on these points even when the disclosure is negative.

| | |
|---|---|
| Study description | |
| Research sample | |
| Sampling strategy | |
| Data collection | |
| Timing | |
| Data exclusions | |
| Non-participation | |
| Randomization | |

# Ecological, evolutionary & environmental sciences study design

All studies must disclose on these points even when the disclosure is negative.

| | |
|---|---|
| Study description | |
| Research sample | |
| Sampling strategy | |
| Data collection | |
| Timing and spatial scale | |
| Data exclusions | |
| Reproducibility | |
| Randomization | |
| Blinding | |

Did the study involve field work?  ☐ Yes  ☐ No

## Field work, collection and transport

| | |
|---|---|
| Field conditions | |
| Location | |
| Access & import/export | |
| Disturbance | |

# Reporting for specific materials, systems and methods

We require information from authors about some types of materials, experimental systems and methods used in many studies. Here, indicate whether each material, system or method listed is relevant to your study. If you are not sure if a list item applies to your research, read the appropriate section before selecting a response.

### Materials & experimental systems

| n/a | Involved in the study |
|---|---|
| ☐ | ☐ Antibodies |
| ☒ | ☐ Eukaryotic cell lines |
| ☒ | ☐ Palaeontology and archaeology |
| ☒ | ☐ Animals and other organisms |
| ☒ | ☐ Clinical data |
| ☒ | ☐ Dual use research of concern |
| ☒ | ☐ Plants |

### Methods

| n/a | Involved in the study |
|---|---|
| ☐ | ☒ ChIP-seq |
| ☒ | ☐ Flow cytometry |
| ☒ | ☐ MRI-based neuroimaging |

## Antibodies

| | |
|---|---|
| Antibodies used | Custom antibodies against Aeropyrum pernix RAD5O were produced in rabbits from purified recombinant protein. |
| Validation | Antibodies' specificity was tested in western-blot (1/100 dilution of primary antibody) with purified recombinant Aeropyrum pernix RADSO protein and whole cell extracts. |

# Eukaryotic cell lines

Policy information about cell lines and Sex and Gender in Research

Cell line source(s)

Authentication

Mycoplasma contamination

Commonly misidentified lines
(See ICLAC register)

# Palaeontology and Archaeology

Specimen provenance

Specimen deposition

Dating methods

☐ Tick this box to confirm that the raw and calibrated dates are available in the paper or in Supplementary Information.

Ethics oversight

Note that full information on the approval of the study protocol must also be provided in the manuscript.

# Animals and other research organisms

Policy information about studies involving animals; ARRIVE guidelines recommended for reporting animal research, and Sex and Gender in Research

Laboratory animals

Wild animals

Reporting on sex

Field-collected samples

Ethics oversight

Note that full information on the approval of the study protocol must also be provided in the manuscript.

# Clinical data

Policy information about clinical studies

All manuscripts should comply with the ICMJE guidelines for publication of clinical research and a completed CONSORT checklist must be included with all submissions.

Clinical trial registration

Study protocol

Data collection

Outcomes

# Dual use research of concern

Policy information about dual use research of concern

## Hazards

Could the accidental, deliberate or reckless misuse of agents or technologies generated in the work, or the application of information presented in the manuscript, pose a threat to:

| No | Yes | |
|----|-----|---|
| x | ☐ | Public health |
| x | ☐ | National security |
| x | ☐ | Crops and/or livestock |
| x | ☐ | Ecosystems |
| x | ☐ | Any other significant area |

## Experiments of concern

Does the work involve any of these experiments of concern:

| No | Yes | |
|----|-----|---|
| x | ☐ | Demonstrate how to render a vaccine ineffective |
| x | ☐ | Confer resistance to therapeutically useful antibiotics or antiviral agents |
| x | ☐ | Enhance the virulence of a pathogen or render a nonpathogen virulent |
| x | ☐ | Increase transmissibility of a pathogen |
| x | ☐ | Alter the host range of a pathogen |
| x | ☐ | Enable evasion of diagnostic/detection modalities |
| x | ☐ | Enable the weaponization of a biological agent or toxin |
| x | ☐ | Any other potentially harmful combination of experiments and agents |

# Plants

| Seed stocks | |
|---|---|
| Novel plant genotypes | |
| | |
| Authentication | |

# ChIP-seq

## Data deposition

☑ Confirm that both raw and final processed data have been deposited in a public database such as GEO.

☐ Confirm that you have deposited or provided access to graph files (e.g. BED files) for the called peaks.

| Data access links
*May remain private before publication.* | NIH SRA BioProject ID PRJNA1027590: http://www.ncbi.nlm.nih.gov/bioproject/1027590 |
|---|---|
| Files in database submission | |
| Genome browser session
(e.g. UCSC) | Not applicable, no peaks called - non-seqeunce specific interactions with DNA |

## Methodology

| Replicates | 2 |
|---|---|
| Sequencing depth | Paired-end read sequencing |
| Antibodies | Custon anti-Ap. pernix RAD50 antisera |
| Peak calling parameters | No peaks called |
| Data quality | FastQC |

| Software | FastQC, SeqMonk. |
|---|---|

# Flow Cytometry

## Plots

Confirm that:

☐ The axis labels state the marker and fluorochrome used (e.g. CD4-FITC).

☐ The axis scales are clearly visible. Include numbers along axes only for bottom left plot of group (a 'group' is an analysis of identical markers).

☐ All plots are contour plots with outliers or pseudocolor plots.

☐ A numerical value for number of cells or percentage (with statistics) is provided.

## Methodology

| Sample preparation | |
|---|---|
| Instrument | |
| Software | |
| Cell population abundance | |
| Gating strategy | |

☐ Tick this box to confirm that a figure exemplifying the gating strategy is provided in the Supplementary Information.

# Magnetic resonance imaging

## Experimental design

| Design type | |
|---|---|
| Design specifications | |
| Behavioral performance measures | |

| Imaging type(s) | |
|---|---|
| Field strength | |
| Sequence & imaging parameters | |
| Area of acquisition | |

Diffusion MRI      ☐ Used      ☐ Not used

## Preprocessing

| Preprocessing software | |
|---|---|
| Normalization | |
| Normalization template | |
| Noise and artifact removal | |
| Volume censoring | |

## Statistical modeling & inference

| Model type and settings | |
|---|---|
| Effect(s) tested | |

Specify type of analysis: ☐ Whole brain   ☐ ROI-based   ☐ Both

Statistic type for inference

(See Eklund et al. 2016)

Correction

## Models & analysis

| n/a | Involved in the study |
|---|---|
| ☐ | ☐ Functional and/or effective connectivity |
| ☐ | ☐ Graph analysis |
| ☐ | ☐ Multivariate modeling or predictive analysis |

Functional and/or effective connectivity

Graph analysis

Multivariate modeling and predictive analysis

