## [Peer Review File · Nature Microbiology]

Peer Review Information

Journal: Nature Microbiology

Manuscript Title: Chromosome architecture in an archaeal species that naturally lacks Structural Maintenance of Chromosomes proteins.

Corresponding author name(s): Stephen Bell

Reviewer Comments & Decisions:

Decision Letter, initial version:
--

Message: 4th April 2023

Dear Dr Bell,

Thank you for your patience while your manuscript "Chromosome architecture in an archaeal species that naturally lacks Structural Maintenance of Chromosomes proteins." was under peer-review at Nature Microbiology. It has now been seen by 3 referees, whose expertise and comments you will find at the end of this email. Overall, the referees are positive about the topic and the potential of the work, however they have raised a number of concerns that will need to be addressed before we can consider publication of the work in Nature Microbiology.

As you will see in the comments, Reviewers #1 and #2 have numerous more minor concerns, the most important being the point from Reviewer #2 about chloramphenicol, which will need to be addressed. Reviewer #3 was the most critical, and suggested overall that the revised version should add a more robust comparison between *A. pernix* and *Sulfolobus*, and we agree that this will increase the broader relevance and impact of your work.

Should further experimental data allow you to address these criticisms, we would be happy to look at a revised manuscript.

We strongly support public availability of data. Please place the data used in your paper

into a public data repository, if one exists, or alternatively, present the data as Source Data or Supplementary Information. If data can only be shared on request, please explain why in your Data Availability Statement, and also in the correspondence with your editor. For some data types, deposition in a public repository is mandatory - more information on our data deposition policies and available repositories can be found at <https://www.nature.com/nature-research/editorial-policies/reporting-standards#availability-of-data>.

Please include a data availability statement as a separate section after Methods but before references, under the heading "Data Availability". This section should inform readers about the availability of the data used to support the conclusions of your study. This information includes accession codes to public repositories (data banks for protein, DNA or RNA sequences, microarray, proteomics data etc...), references to source data published alongside the paper, unique identifiers such as URLs to data repository entries, or data set DOIs, and any other statement about data availability. At a minimum, you should include the following statement: "The data that support the findings of this study are available from the corresponding author upon request", mentioning any restrictions on availability. If DOIs are provided, we also strongly encourage including these in the Reference list (authors, title, publisher (repository name), identifier, year). For more guidance on how to write this section please see: <http://www.nature.com/authors/policies/data/data-availability-statements-data-citations.pdf>

* If you have not done so already we suggest that you begin to revise your manuscript so that it conforms to our Article format instructions at <http://www.nature.com/nmicrobiol/info/final-submission>. Refer also to any guidelines provided in this letter.

[redacted]

Note: This url links to your confidential homepage and associated information about manuscripts you may have submitted or be reviewing for us. If you wish to forward this e-mail to co-authors, please delete this link to your homepage first.

Nature Microbiology is committed to improving transparency in authorship. As part of our efforts in this direction, we are now requesting that all authors identified as 'corresponding author' on published papers create and link their Open Researcher and Contributor Identifier (ORCID) with their account on the Manuscript Tracking System (MTS), prior to acceptance. This applies to primary research papers only. ORCID helps the scientific community achieve unambiguous attribution of all scholarly contributions. You can create and link your ORCID from the home page of the MTS by clicking on 'Modify my Springer Nature account'. For more information please visit www.springernature.com/orcid.

If you wish to submit a suitably revised manuscript we would hope to receive it within 6 months. If you cannot send it within this time, please let us know. We will be happy to consider your revision, even if a similar study has been accepted for publication at Nature Microbiology or published elsewhere (up to a maximum of 6 months).

Yours sincerely,

Kyle

Dr. Kyle Frischkorn
(he/him/his)
Senior Editor, Nature Microbiology
Nature Portfolio

Reviewer Expertise:

Referee #1: chromosome architecture
Referee #2: archaeal genetics
Referee #3: genome organization

Reviewer Comments:

Reviewer #1 (Remarks to the Author):

In the manuscript entitled « Chromosome architecture in an archeal species that naturally lacks Structural Maintenance of Chromosome proteins », Badel and Bell wish to analyze chromosome conformation in Desulforococcales model species *Aeropyrum pernix*. The interesting results include the observation of local CID domain around the chromosome, a dependance on transcription for its strength, an additional higher order organization involving a self-interacting domain and loops which reconfigurations occur upon treatment with Actinomycin D that partially affects transcription. The experiments are well performed and this report contains several important and interesting results. However, the inhibition of translation and its effects of chromosome conformation are not obvious and several other raised points could be better explained.

Major points:

- It might be useful to include a phylogenic tree as that published by Takemata et al. (2019) to show the distribution of SMC and SMC-like proteins in Archaea. In addition to Desulforococcales, other Crenarchaeota in the TACK superphylum are apparently devoid of SMC-like proteins; this should be discussed. Also, why *Aeropyrum pernix* was chosen could be explained. The tree would help to follow the discussion on the evolution of SMC in Archaea.
- Figure 1: The MFA pattern in stationary phase either indicates that replication is still ongoing or that the population contains a mixture of cells with replication arrested at different loci. Since chromosome conformation is not addressed for cells in stationary phase, the replication analysis in stationary phase can be moved to Supplementary Figures.
- Figure 2B and Figure 4E: Heatmap of distance-normalized contact score is probably not the best way to show CID organization. The directionality index (DI) analysis performed (Figure 2C) revealed indirectly the presence of 19 CIDs. Among others, a direct method to visualize CIDs has been published recently (Computational Tools for the Multiscale Analysis of Hi-C Data in Bacterial Chromosomes. *Methods Mol Biol.* (2022) 2301:197-207.)
- The Chromosight method detects 171 loops, some of them involving very distant anchors and shared anchor points. The authors should comment on loops detected at a higher observed/expected ratio, i.e. in less transcribed genes, in the provirus APSV1 and in the CRISPR locus. The shapes of the signals in Figure C are not discussed.
- The effect of Cm on chromosome conformation seems to be weak and as stated by the authors, the changes observed were weak "possibly owing to a modest translation disruption". This part of analysis may consequently be either removed from the manuscript or shortened and transferred to Supplementary Figures. The Contact maps in the presence of Actinomycin D and Cloramphenicol are not shown.

Other points:

- Figure 4F is not described in the Figure legend.
- Figure 5H is missing; it should be Figure 5I
- Figure S10 should be Figure S7

Reviewer #2 (Remarks to the Author):

The manuscript by Badel and Bell describes a comprehensive investigation of the chromosome architecture in the archaeal species *Aeropyrum pernix*. The authors have chosen to study this species because it lacks the genes for SMC-superfamily proteins, with

the sole exception of Rad50.

The techniques used by the authors include MFA-seq to study DNA replication origins, a variety of chromosome conformation capture experiments to examine chromosome architecture in *Aeropyrum pernix*, which are complemented by bioinformatic analysis to detect loops, and RNA-seq to study transcription. The experiments are described succinctly and the data is reported carefully. The authors are to be commended on their comprehensive description of the methods and their elegant data visualisation.

The findings are original and provide valuable insights into the evolution of chromosome architecture in archaea, specifically the *Sulfolobus* lineage. This manuscript will be of immediate and widespread interest, specifically to those working on chromosome architecture in all domains of life. The work will also be of interest throughout the archaeal community, in particular those researchers concerned with evolution of the last common ancestor of archaea and eukaryotes. For example, the suggestion that the *clsN* gene has been acquired from an extrachromosomal element is intriguing and worthy of further study.

My recommendation is for publication in *Nature Microbiology*, subject to addressing the comments below.

Major

Page 2 lines 39-40. The authors state that organisms in the Desulfurococcales phylum of Crenarchaeota lack the genes encoding SMC-superfamily proteins, with the exception of Rad50. A reference is missing here. If the authors have carried out this analysis themselves, they should provide the data.

Page 2 lines 71-72, and page 4 lines 138-141. The enrichment of Rad50 at transcriptionally active loci may be due to its role in the repair of DNA double-strand breaks, which are likely to be formed by the processing of R-loops at stalled/aborted transcription events. Unless the authors disagree strongly with this suggestion, perhaps it should be mentioned?

Page 3 lines 100 onwards. I am puzzled by the decision to use chloramphenicol. While this translation inhibitor has been shown to work in some archaea such as *Sulfolobus acidocaldarius* (<https://pubmed.ncbi.nlm.nih.gov/8002603/>), it is not effective in others such as *Haloferax mediterranei* (<https://pubmed.ncbi.nlm.nih.gov/14654696/>). Has chloramphenicol been tested in *Aeropyrum pernix*? Furthermore, anisomycin has been shown to be an effective translation inhibitor in several archaeal species (<https://pubmed.ncbi.nlm.nih.gov/18455733/>), so why not use anisomycin instead?

Page 3 lines 104-105. The authors claim that "To the best of our knowledge, this is the first study to analyze the impact of translation on chromosome conformation in archaea". This is not strictly speaking true. A previous study (<https://pubmed.ncbi.nlm.nih.gov/23145964/>) employed the translation inhibitor anisomycin to study nucleoid (chromosome) compaction in *Haloferax volcanii*. Admittedly, the study described in the manuscript currently under review is significant more comprehensive, but nevertheless this study by Delmas et al should be cited. There are further parallels with the manuscript currently under review, since Delmas et al found that Rad50 plays a critical role in nucleoid compaction in response to the translation inhibitor

anisomycin.

Page 4 line 124. The authors report that DMSO leads to transcription induction of a number of loci. Are these loci concerned with aerobic and/or anaerobic respiration? DMSO can be used as terminal electron acceptor in anaerobic growth, so is this likely to be the case here? (<https://pubmed.ncbi.nlm.nih.gov/15716436/>
<https://pubmed.ncbi.nlm.nih.gov/414686/>)

Minor

Page 1 line 12. CID is used but not (explicitly) defined.

Page 2 line 60. PCA is spelled incorrectly throughout, it should be Principal Component Analysis (not Principle).

Page 3 line 106 and line 116. One line 106 it is stated that "Upon transcription inhibition, long-range contacts decreased..." and in line 116 it says "Upon transcription inhibition, long-range contacts increased overall...". Is the former (line 106) a typographical error and should read "Upon translation inhibition, long-range contacts decreased...?"

Page 5 line 190. The authors state that cells were resuspended and that "400 L [sic] of the suspension was centrifuged". I appreciate that such experiments may require a lot of material but this seems excessive.

Page 7 lines 273-274, and Figure 1. I agree that accounting for GC bias is best way to normalise with Illumina MFAseq but nonetheless, it is notable that the origins are still visible in the stationary phase sample in Figure 1B. Why might this be?

This review has been written by Thorsten Allers

Reviewer #3 (Remarks to the Author):

This research article by Badel and Bell describes chromosome architecture in *Aeropyrum pernix* – an archaeal model organism of the Desulfurococcales phylum that naturally lack the Structural Maintenance of Chromosomes (SMC) proteins. The study was meant to understand how organisms may organize chromosomes in the absence of SMC and SMC-like proteins that, up until now, have been observed to be essential for chromosome structuring. They find that in this model organism the chromosome is organized into local domains (chromosome interaction domains – CIDs) similar to other prokaryotes, and that global changes in transcription and translation, realized in this study by treatment with antibiotics, influence this organisation. The authors also show that higher-order chromosome organization is determined by transcription level and involves segregation into HEID and ROC (high gene expression self-interacting domain and rest of chromosome). The authors speculate that the organization of HEID as a domain segregated based on transcriptional activity is an evolutionary antecedent of A/B-compartmentalisation observed in *Sulfolobus*.

The motivation of the study outlined in the abstract is exciting, but the manuscript falls short of these expectations especially from a biological and evolutionary perspective. The

manuscript is merely a descriptive text of the organization of the *A. pernix* chromosome, and the conclusions only describe the data. Comparisons between the chromosome organization of *A. pernix* and *Sulfolobus* as alluded to in the abstract and introduction are minimal and the biological significance of their findings have not been discussed. This leads to the author's suggestion in the abstract and the discussion that the organization of the *A. pernix* chromosome is an evolutionary antecedent of *Sulfolobus* seem so speculative that the suggestion does not merit the importance that has been placed on it. The manuscript also falls short of providing insight on the role of SMCs.

Nevertheless, the motivation of this work is exciting. It has the potential to be extremely useful for the fields of archaeal biology and chromatin biology. Therefore, I suggest major revisions of the manuscript that focus on drawing parallels and highlighting differences between the organization of the chromosome of *A. pernix* and *Sulfolobales*, and where applicable other archaea, for every individual test/experiment that is described (for instance, the distribution of highly transcribed genes in the chromosome, general trends of loop distribution, etc.). Where possible, the authors should describe the evolutionary and biological significance of the features of chromosome organization in *A. pernix*, and of similarities and differences between chromosome organization in *A. pernix* and other archaea. The selling point of this work is its potential to provide insight into the role of SMCs. The authors should expand their discussion in this area.

General comments:

- 1) All figures lack sufficient explanation in the main text and in the figure legends. Important features of the graphs that lead the authors to their conclusions are sometimes not obvious or highlighted, and, as such, leave it up to the reader to speculate how the conclusions were made. The figures have not been used to properly supplement the text and the figure legends are minimalistic and do not sufficiently describe the figures. Often several panels of a figure are cross-referenced for a conclusion without a proper description.
- 2) The authors have not justified the use of the two-sided Wilcoxon test that is frequently used in this manuscript to compare two populations.
- 3) At the beginning of the manuscript, the authors use 'local domains' and 'local interaction domains' to describe what are generally referred to as 'chromosome interaction domains' (CIDs) almost as if actively avoiding the use of CIDs as a terminology to describe chromosome organisation in *A. pernix*. However, in the discussion, the authors resort to using CIDs. Regardless of what terminology the authors choose to use, it should be standardized. In that light, could the authors also correct their abstract in line 12 '... local domains (CIDs)...'? CIDs is not an abbreviation for local domains.

Specific comments:

- 1) In the introduction, lines 38-42, the evolutionary divergence between *Sulfolobus* and *Aeropyrum* should be explained with a (supplementary) figure of an evolutionary tree marking the positions of these organisms and other model organisms mentioned in the paper. The figure should also highlight/mark where SMCs and SMC-like proteins have been lost or gained.
- 2) Figures 1A, 1B: The normalized read count along the chromosome shown by the red and blue lines indicates that the difference in the relative copy number of the origin compared to the less replicated regions of the chromosome is larger in stationary phase than in exponential phase. What does this suggest? At exactly what points on the *Aeropyrum pernix* growth curve were the exponential and stationary phase samples collected?

- 3) Figures 1D-1G; Lines 48-52: In profiling the transcription of the genes across the chromosome during exponential and stationary phases to identify highly expressed regions, were the differences in the copy numbers of the loci as represented in Figures 1A and 1B accounted for?
- 4) Lines 57-58: 'Despite our best efforts, no individual distinctive feature could be identified at CID borders.' What was searched for? How? What were the results of these searches? The information would be useful to have in the supplementary.
- 5) Figure S5: Why are these graphs in the supplementary while the results shown in these are mentioned in the text as main findings? For three of the graphs shown in Figure S5, a mention of the statistical test used to compute the correlation coefficient and p value is missing.
- 6) Several figure legends say, 'The frequency of the actual value in the random simulation is indicated as the p-value.' Please justify/explain.
- 7) Figure 3A: Some loops have been identified in regions where 'stripes' cross. Could these loops be artefacts of normalization? How many loop anchor sites correspond to areas of local depletion in read depth? Preparing this library with a restriction enzyme having a different digestion site and applying the loop detection software for the new contact matrices would be useful in verifying which loops are real and which ones are artefacts.
- 8) The paragraph from line 81 to line 90 is about chromosome looping in *A. pernix* and the observation that loops are anchored between loci with similar transcriptional level. How are these loops anchored?
- 9) Lines 86-88: '... raising the possibility of the provirus using transcriptional coupling.' What does transcriptional coupling mean? Explaining this hypothesis in more detail would be beneficial.
- 10) Lines 98-100: '... 91/1753 genes were significantly induced ($p < 0.01$), including... the chromatin protein Cren7.' By what factor was this gene induced? Are there any features of the chromosome contact map of the ActD treated cells that may be attributed to Cren7-mediated chromosome structuring?
- 11) Figure 4E: The pattern of chromosome contacts does not fit with 'squares along the main diagonal' that is typical of CIDs. Hence, the CIDs marked in Figure 4E (with Directionality Index measurements) are not convincing. Does the chromosome contact map show 'typical' CIDs? Is Directionality Index really a reliable way of assigning short range contacts in this map? Should a different perspective/model be used to describe/explain the short-range contacts in this organism?
- 12) In lines 119-121 the authors say that the location of HEID' correlates with the location of ActD resistant transcription. Looking at the data presented in Figure 5F, I disagree with this statement. Several ActD resistant genes are not part of HEID'. What percentage of all the ActD resistant genes with a p-value < 0.01 lie within the HEID'? What are Figures 5G and 5H meant to show?
- 13) Lines 488-498 say that the aggregate contact maps show median values. This is in contrast to lines 232-233 (Methods section, Aggregate contact maps) that say that an average matrix is represented.
- 14) Figure S7: Those are not growth curves. More measurements are required between inoculation and the treatment period. The '30min treatment period' for ActD and DMSO is longer than for chloramphenicol and ethanol.

Minor comments:

- 1) Lines 47-48: Figures 1C-1H are referenced with regards to gene conservation. The only figures that show data relating to gene conservation are Figures 1C and 1H.
- 2) Figure 1C: What do the letters A, AA, AAP, AAPD mean?

- 3) Figure 1F: The Pearson correlation coefficient has not been provided. A reference to the figure is missing in line 51.
- 4) Figure 1G; Lines 51-52: The p-value provided in the main text and that in the figure are not the same. A reference to Figure 1G is missing in line 52.
- 5) Figure 2B: The thick line used to mark the CIDs seems to obscure the features that may be used to visually distinguish CIDs. Could you use a thinner line? Visually, I can only confidently agree with CIDs 7 and 15 from the left.
- 6) Figure 2C: This figure looks weird: the directionality indices for the loci are overlapping. The reference for calculating directionality indices in the Methods section measures directionality preferences (is it the same?) and represents the data differently. Either the representation should be standardized, or the analysis performed for this paper should be described in more detail in the Methods section.
- 7) The figures have not been referenced sequentially throughout the text. Figures S2 and S3 are not referenced in the main text. Figure S10 (line 103) is not there.
- 8) Figure 2G: HEID and ROC terminologies are used in the figure before they are introduced in the main text.
- 9) Figure 4D-right: The triangles marking the transcription-related genes are difficult to distinguish. Marking these genes using a color that contrasts the black and grey of the figure will highlight these better.
- 10) Line 102-105: Here, it looks as if the authors are missing a reliable test to verify the effect of Chloramphenicol. Perhaps SDS-PAGE?
- 11) In lines 102-104 the authors say that growth is retarded in cultures treated with chloramphenicol. In lines 112-114 they suggest that chloramphenicol treatment may modestly disrupt translation. These sentences are contradictory.
- 12) Line 122: Perhaps the authors want to refer to Figures 5I and 5J?
- 13) The Materials and Methods section has multiple errors. For instance, the names of the chemicals in the materials and methods are not written with the proper subscripts, the space between numbers and units is missing, for instance, 0.5M instead of 0.5 M, in some places the capital letter 'O' has been substituted by the number '0', mentions of optical density (OD) do not have an associated wavelength, on occasion 'u' has been used instead of 'μ', etc. Please go over the Materials and Methods section again and correct these.
- 14) Line 210-211: Why were the genomic coordinates redefined? Why were the original coordinates not used?
- 15) The 3C-Seq experiments were done in replicate. Show the individual contact maps in the supplementary information.
- 16) Line 246-247: The sentence is incorrect.
- 17) Lines 270-271: '... Tris, pH=8. DNA. DNA libraries...' There seems to be an error here.
- 18) Line 276: '...calculated as observed-(theoretical-average) with...' Is this supposed to be a formula? It is not written so clearly.
- 19) Lines 313-314: Is the composition of TBS-TT buffer correct?
- 20) Lines 388-390: This reference has not been formatted properly.
- 21) Split the supplementary figures into panels, and check the figure legends: Figure S8 is not split into panels as has been described in the figure legend. There is no panel J in Figure S9.

Author Rebuttal to Initial comments**Reviewer #1:**

Major points:

- It might be useful to include a phylogenetic tree as that published by Takemata et al. (2019) to show the distribution of SMC and SMC-like proteins in Archaea. In addition to Desulfurococcales, other Crenarchaeota in the TACK superphylum are apparently devoid of SMC-like proteins; this should be discussed. Also, why *Aeropyrum pernix* was chosen could be explained. The tree would help to follow the discussion on the evolution of SMC in Archaea.

Sure, we've added a new Figure S1.

- Figure 1: The MFA pattern in stationary phase either indicates that replication is still ongoing or that the population contains a mixture of cells with replication arrested at different loci. Since chromosome conformation is not addressed for cells in stationary phase, the replication analysis in stationary phase can be moved to Supplementary Figures.

The panels corresponding to cells in stationary phase were moved in supplementary figure S4 as suggested. We have also expanded our discussion of this phenomenon in the main text

“We note that MFA analyses performed in stationary phase reveal a marker distribution similar to exponentially growing cells, indeed the amplitude of the peaks corresponding to replication initiation is actually greater than in exponentially growing cells. This striking observation is in agreement with previously published flow cytometry data that revealed an elevated GF1/ early S-phase population in stationary phase *A. pernix* cells. (Lundgren et al., 2008, <https://doi.org/10.1128/jb.00330-08>). »

- Figure 2B and Figure 4E: Heatmap of distance-normalized contact score is probably not the best way to show CID organization. The directionality index (DI) analysis performed (Figure 2C) revealed indirectly the presence of 19 CIDs. Among others, a direct method to visualize CIDs has been published recently (Computational Tools for the Multiscale Analysis of Hi-C Data in Bacterial Chromosomes. *Methods Mol Biol.* (2022) 2301:197-207.)

We implemented the suggested method to visualize CID organization (see figure below). It does not work as well for *Aeropyrum pernix* as for *P. aeruginosa*, possibly because of the higher order organization present in *Aeropyrum pernix* that seems to dominate the frontier signal. We therefore decided to keep the original representation method for the CIDs – this will also facilitate comparisons with the similarly analyzed *Sulfolobus* data that we have published.

Figure. Frontiers signal for *Aeropyrum pernix* obtained using the same method as in https://doi.org/10.1007/978-1-0716-1390-0_10. Axis are scaled in bins of 3 kb.

- The Chromosight method detects 171 loops, some of them involving very distant anchors and shared anchor points. The authors should comment on loops detected at a higher observed/expected ratio, i.e. in less transcribed genes, in the provirus APSV1 and in the CRISPR locus. The shapes of the signals in Figure C are not discussed.

Comments were added on loops anchored in less transcribed genes, the provirus APSV1 and the CRISPR locus.

- The effect of Cm on chromosome conformation seems to be weak and as stated by the authors, the changes observed were weak “possibly owing to a modest translation disruption”. This part of analysis may consequently be either removed from the manuscript or shortened and transferred to Supplementary Figures. The Contact maps in the presence of Actinomycin D and Chloramphenicol are not shown.

The comprehensive analysis of chromosome conformation changes due to translation disruption presented in the manuscript is novel for Archaea. We therefore decided to keep it in the main part, despite the possible modest translation disruption.

A supplementary figure was also added with all the replicates contact maps, including Actinomycin D and Chloramphenicol, as supplementary figure S6

Other points:

- Figure 4F is not described in the Figure legend.

This omission was corrected.

- Figure 5H is missing; it should be Figure 5I

Figure 5H is not missing, it is the second violin plot next to the 5G panel.

- Figure S10 should be Figure S7

The typo was corrected.

Reviewer #2 (Remarks to the Author):**Major**

Page 2 lines 39-40. The authors state that organisms in the Desulfurococcales phylum of Crenarchaeota lack the genes encoding SMC-superfamily proteins, with the exception of Rad50. A reference is missing here. If the authors have carried out this analysis themselves, they should provide the data.

We've added a figure (Figure S1) to this effect and also a reference to recent analyses of SMC superfamily proteins in Archaea.

Page 2 lines 71-72, and page 4 lines 138-141. The enrichment of Rad50 at transcriptionally active loci may be due to its role in the repair of DNA double-strand breaks, which are likely to be formed by the processing of R-loops at stalled/aborted transcription events. Unless the authors disagree strongly with this suggestion, perhaps it should be mentioned?

Thank you for the suggestion. This comment was added.

Page 3 lines 100 onwards. I am puzzled by the decision to use chloramphenicol. While this translation inhibitor has been shown to work in some archaea such as *Sulfolobus acidocaldarius* (<https://pubmed.ncbi.nlm.nih.gov/8002603/>), it is not effective in others such as *Haloferax mediterranei* (<https://pubmed.ncbi.nlm.nih.gov/14654696/>). Has chloramphenicol been tested in *Aeropyrum pernix*? Furthermore, anisomycin has been shown to be an effective translation inhibitor in several archaeal species (<https://pubmed.ncbi.nlm.nih.gov/18455733/>), so why not use anisomycin instead?

The drug used to inhibit translation in *Aeropyrum pernix* was chosen among drugs that were known to have an effect on *Sulfolobus acidocaldarius*, the model strain that is the more closely related to *A. pernix*. Both chloramphenicol and puromycin were tested on *Aeropyrum pernix*. Chloramphenicol was selected because it lead to a more reliable growth defect.

Note that anisomycin was reported not to have an effect on translation in *Sulfolobus acidocaldarius* (DOI: 10.1128/jb.176.24.7744-7747.1994)

Page 3 lines 104-105. The authors claim that "To the best of our knowledge, this is the first study to analyze the impact of translation on chromosome conformation in archaea". This is not strictly speaking true. A previous study (<https://pubmed.ncbi.nlm.nih.gov/23145964/>) employed the translation inhibitor anisomycin to study nucleoid (chromosome) compaction in *Haloferax volcanii*. Admittedly, the study described in the manuscript currently under review is significant more comprehensive, but nevertheless this study by Delmas et al should be cited. There are further parallels with the manuscript currently under review, since Delmas et al found that Rad50 plays a critical role in nucleoid compaction in response to the translation inhibitor anisomycin.

Thanks for drawing our attention to this – we've removed the claim of novelty and added a comment regarding the Haloferax data.

Page 4 line 124. The authors report that DMSO leads to transcription induction of a number of loci. Are these loci concerned with aerobic and/or anaerobic respiration? DMSO can be used as terminal electron acceptor in anaerobic growth, so is this likely to be the case here?

(<https://pubmed.ncbi.nlm.nih.gov/15716436/> <https://pubmed.ncbi.nlm.nih.gov/414686/>)

Among the most DMSO-induced genes are two genes annotated as tetrathionate and polysulfide reductase. Both reductase belong to the DMSO reductase family and are involved in anaerobic respiration (<https://doi.org/10.1038/s41598-020-67892-9>). However, a gene coding for an arsenate reductase belonging to the same DMSO reductase family was down regulated by DMSO and only one gene coding for proteins related to aerobic respiration (cytochrome c biogenesis protein Ccsa) was downregulated. So it seems that *Aeropyrum pernix* is indeed taking advantage of the DMSO as a terminal electron acceptor as suggested but without completing switching its energy metabolism.

Minor

Page 1 line 12. CID is used but not (explicitly) defined.

Corrected.

Page 2 line 60. PCA is spelled incorrectly throughout, it should be Principal Component Analysis (not Principle).

Corrected.

Page 3 line 106 and line 116. One line 106 it is stated that "Upon transcription inhibition, long-range contacts decreased..." and in line 116 it says "Upon transcription inhibition, long-range contacts increased overall...". Is the former (line 106) a typographical error and should read "Upon translation inhibition, long-range contacts decreased..."?

Corrected.

Page 5 line 190. The authors state that cells were resuspended and that "400 L [sic] of the suspension was centrifuged". I appreciate that such experiments may require a lot of material but this seems excessive.

Thanks for noticing this immense typo. It was corrected.

Page 7 lines 273-274, and Figure 1. I agree that accounting for GC bias is best way to normalise with Illumina MFAseq but nonetheless, it is notable that the origins are still visible in the stationary phase sample in Figure 1B. Why might this be?

Note that the stationary phase data was moved to supplementary. Perhaps the apparent stationary phase observed by absorption at OD600 nm reflects balanced growth and death rates in the culture with ongoing replication occurring while a proportion of the culture is dying. We have expanded our discussion of this observation in the main text “We note that MFA analyses performed in stationary phase reveal a marker distribution similar to exponentially growing cells, indeed the amplitude of the peaks corresponding to replication initiation is actually greater than in exponentially growing cells. This striking observation is in agreement with previously published flow cytometry data that revealed an elevated GF1/ early S-phase population in stationary phase A. *pernix* cells. (Lungren et al., 2008, <https://doi.org/10.1128/jb.00330-08>). »

Reviewer #3:

The motivation of the study outlined in the abstract is exciting, but the manuscript falls short of these expectations especially from a biological and evolutionary perspective. The manuscript is merely a descriptive text of the organization of the *A. pernix* chromosome, and the conclusions only describe the data. Comparisons between the chromosome organization of *A. pernix* and *Sulfolobus* as alluded to in the abstract and introduction are minimal and the biological significance of their findings have not been discussed. This leads to the author's suggestion in the abstract and the discussion that the organization of the *A. pernix* chromosome is an evolutionary antecedent of *Sulfolobus* seem so speculative that the suggestion does not merit the importance that has been placed on it. The manuscript also falls short of providing insight on the role of SMCs.

General comments:

1) All figures lack sufficient explanation in the main text and in the figure legends. Important features of the graphs that lead the authors to their conclusions are sometimes not obvious or highlighted, and, as such, leave it up to the reader to speculate how the conclusions were made. The figures have not been used to properly supplement the text and the figure legends are minimalistic and do not sufficiently describe the figures. Often several panels of a figure are cross-referenced for a conclusion without a proper description.

We have attempted to clarify and improve the figure legends and referencing to figures throughout.

2) The authors have not justified the use of the two-sided Wilcoxon test that is frequently used in this manuscript to compare two populations.

The justification was added in the material and method section.

3) At the beginning of the manuscript, the authors use 'local domains' and 'local interaction domains' to describe what are generally referred to as 'chromosome interaction domains' (CIDs) almost as if actively avoiding the use of CIDs as a terminology to describe chromosome organisation in *A. pernix*. However, in the discussion, the authors resort to using CIDs. Regardless of what terminology the authors choose to use, it should be standardized. In that light, could the authors also correct their abstract in line 12 '... local domains (CIDs)...'? CIDs is not an abbreviation for local domains.

Our apologies, we at no point intended to avoid the use of the CID nomenclature – we simply sought to describe the features of the "local domains" then make use of CID throughout. We have carefully gone through the paper and have, hopefully, clarified this issue.

Specific comments:

- 1) In the introduction, lines 38-42, the evolutionary divergence between *Sulfolobus* and *Aeropyrum* should be explained with a (supplementary) figure of an evolutionary tree marking the positions of these organisms and other model organisms mentioned in the paper. The figure should also highlight/mark where SMCs and SMC-like proteins have been lost or gained.

Thanks for the suggestion – we have added the suggested figure.

- 2) Figures 1A, 1B: The normalized read count along the chromosome shown by the red and blue lines indicates that the difference in the relative copy number of the origin compared to the less replicated regions of the chromosome is larger in stationary phase than in exponential phase. What does this suggest? At exactly what points on the *Aeropyrum pernix* growth curve were the exponential and stationary phase samples collected?

Note that the stationary phase data was moved to supplementary. Perhaps the apparent stationary phase observed by absorption at OD600 nm reflects balanced growth and death rates in the culture with ongoing replication occurring while a proportion of the culture is dying.

Sampling time points were added in supplementary figure S2.

- 3) Figures 1D-1G; Lines 48-52: In profiling the transcription of the genes across the chromosome during exponential and stationary phases to identify highly expressed regions, were the differences in the copy numbers of the loci as represented in Figures 1A and 1B accounted for?

The difference in copy number of the loci were not accounted for in calculating RPKSP.

We suspect that you imply that the difference between the transcriptional level of HEID and ROC might be due to a higher copy number of genes in the HEID than the ROC rather than a stronger transcriptional activity in the HEID ; and therefore that a such stronger transcriptional activity could not explain the formation of the HEID. To quickly test for this possibility, we compared the transcriptional level between the HEID normalized by a factor of 9/5 for all genes (the strongest difference of copy number in exponential phase) and the ROC. Such an exaggerated normalization also presented a statistically significant difference ($p < 2.2e-16$). The difference in copy number can therefore not account alone for the higher transcriptional level in the HEID.

- 4) Lines 57-58: 'Despite our best efforts, no individual distinctive feature could be identified at CID borders.' What was searched for? How? What were the results of these searches? The information would be useful to have in the supplementary.

CID boundaries were explored visually to look for common features including genes orientation and transcriptional level. Different parameters were compared by violin plot and Wilcoxon tests between

border bins and non-border bins, including RNA level, GC richness and RAD50 enrichment and evidencing no statistical differences. Permutations tests were also performed on CID localization to test whether loop anchors or certain gene types were enriched at CID boundaries, evidencing no statistical difference. Note that the 3 kb resolution of the analysis might prevent determining a small distinctive feature of the CID.

5) Figure S5: Why are these graphs in the supplementary while the results shown in these are mentioned in the text as main findings? For three of the graphs shown in Figure S5, a mention of the statistical test used to compute the correlation coefficient and p value is missing.

Note that figure S5 is now figure S8

We deemed that the negative result presented in Figure S5 was not worth including as a main figure. The requested mentions of the statistical test were added.

6) Several figure legends say, 'The frequency of the actual value in the random simulation is indicated as the p-value.' Please justify/explain.

This sentence refers to the permutation test section of the material and method. The sentence was modified to make it more explicit: "An empirical p-value is indicated (see material and method)".

7) Figure 3A: Some loops have been identified in regions where 'stripes' cross. Could these loops be artefacts of normalization? How many loop anchor sites correspond to areas of local depletion in read depth? Preparing this library with a restriction enzyme having a different digestion site and applying the loop detection software for the new contact matrices would be useful in verifying which loops are real and which ones are artefacts.

We looked at the raw 3C coverage along the chromosome, i.e. the number of chimeric 3C reads sequenced for each position, and crossed this data with the number of loops anchored at each bin (figure below). First, note that the raw 3C coverage is highly biased by the ongoing replication and that this prevents an analysis over the entire chromosome. However, locally, there is no trend of read depletion for bins where loops are anchored. The bins that have the most loops appear to have a read depth that is in the lower range of the local distribution, but without being an overly extreme value. Additionally, bins with the lowest read depth do not present any loops. Therefore, we are confident that these loops exist and are not just an artifact due to the normalization of poorly covered bins.

Figure. 3C read depth along the chromosome and number of loops anchored in the bin. Bins without any loops are indicated by a grey dot. Bins with loops are colored depending on the number of loops they present.

8) The paragraph from line 81 to line 90 is about chromosome looping in *A. pernix* and the observation that loops are anchored between loci with similar transcriptional level. How are these loops anchored?

This is a very interesting question and one that we are actively pursuing, however, we would respectfully suggest that a full mechanistic dissection of loop anchoring lies beyond the scope of the current manuscript.

9) Lines 86-88: ‘... raising the possibility of the provirus using transcriptional coupling.’ What does transcriptional coupling mean? Explaining this hypothesis in more detail would be beneficial.

The hypothesis was explained in more detail.

10) Lines 98-100: ‘... 91/1753 genes were significantly induced ($p < 0.01$), including... the chromatin protein Cren7.’ By what factor was this gene induced? Are there any features of the chromosome contact map of the ActD treated cells that may be attributed to Cren7-mediated chromosome structuring?

Cren7-mediated chromosome structuring is currently poorly known (<https://doi.org/10.1016/j.jmb.2020.166791>). Cren7 is known to bind preferentially to AT rich regions,

to bend DNA at low protein concentrations and compact DNA at higher concentrations. Cren7 might also be related to the transcription of specific loci in conjunction with other proteins (<https://www.biorxiv.org/content/10.1101/2023.03.24.534125v1.abstract>). A potential effect of Cren7 on chromosome conformation might be mediated by its transcriptional activity. Further studies would be needed to decipher the influence of Cren7 on chromosome structuring.

The RNA level of Cren7 presented a log₂ fold change of 1.47 between the ActD and DMSO conditions. Note that it was shown that protein abundances changes are slower than transcriptional changes in *Sulfolobus*: 15 minutes versus 60 minutes in a heat shock response study (<https://doi.org/10.1101/2022.12.17.520879>). Cren7 protein abundance might not yet be altered when the samples were collected after 30 minutes treatment.

11) Figure 4E: The pattern of chromosome contacts does not fit with ‘squares along the main diagonal’ that is typical of CIDs. Hence, the CIDs marked in Figure 4E (with Directionality Index measurements) are not convincing. Does the chromosome contact map show ‘typical’ CIDs? Is Directionality Index really a reliable way of assigning short range contacts in this map? Should a different perspective/model be used to describe/explain the short-range contacts in this organism?

We used the same method to define CIDs as published in *Sulfolobus islandicus* (Takemata et al. 2021) that present a similar contact map to *A. pernix*. We agree that not all CIDs appear visually to the human eye as squares along the main diagonal in the contact map but (1) the square pattern is more evident in the aggregate map (Figure 2E) and (2) CID appear strongly when looking at the changes in chromosome conformations between ActD or Cam treated samples and their controls (Figure 4). Moreover, we confirmed the location of the border of CIDs with a second index : the insulation score (Figure 2D). All these pieces of evidence prompted us to have confidence in actual existence of CIDs in *A. pernix*.

12) In lines 119-121 the authors say that the location of HEID’ correlates with the location of ActD resistant transcription. Looking at the data presented in Figure 5F, I disagree with this statement. Several ActD resistant genes are not part of HEID’. What percentage of all the ActD resistant genes with a p-value <0.01 lie within the HEID’? What are Figures 5G and 5H meant to show?

An explicit comment to the figures 5G and 5H was added to the text.

Among the 91 genes that are significantly induced upon ActD treatment, 28 (31%) laid in the HEID’, 61 (67%) in ROC and 2 over both HEID’ and ROC. This is slightly biased towards HEID’ compared to the proportion of all genes with 362/1753 (21%) genes in HEID’ and 1368/1753 (78%) in ROC (two sided Fisher test, p-value = 0.0116).

Furthermore, the RNA level and LFC are significantly higher in the HEID’ domain compared to the ROC (Figure 5G and H). All these results indicate that the HEID’ domain is enriched in loci with residual

transcription compared with the ROC and point towards active transcription playing a role in structuring the HEID' domain, in agreement with what was observed in the exponential phase condition. Some ActD-resistant genes are nonetheless present in the ROC as you pointed out. This could be an artefact from the resolution of the analysis and a higher resolution would correctly assign those genes to the HEID' domain or this could suggest that other unknown mechanisms are also involved in the structuration of the HEID' domain.

13) Lines 488-498 say that the aggregate contact maps show median values. This is in contrast to lines 232-233 (Methods section, Aggregate contact maps) that say that an average matrix is represented.

Thank you for noticing this discrepancy, we have corrected it. The average matrix is represented.

14) Figure S7: Those are not growth curves. More measurements are required between inoculation and the treatment period. The '30min treatment period' for ActD and DMSO is longer than for chloramphenicol and ethanol.

Thank you for noticing the inconsistency in the treatment periods. Data were verified and the figure was corrected. The title of the figure was changed.

Minor comments:

1) Lines 47-48: Figures 1C-1H are referenced with regards to gene conservation. The only figures that show data relating to gene conservation are Figures 1C and 1H.

Corrected.

2) Figure 1C: What do the letters A, AA, AAP, AAPD mean?

The letters correspond to different dataset. Their composition is detailed in Table S1. Table S1 was modified to explicit the lettering. Text was modified to make finding this information easier.

3) Figure 1F: The Pearson correlation coefficient has not been provided. A reference to the figure is missing in line 51.

This panel is now 1D. The correlation coefficient was not provided because it is not statistically significant

The reference to the figure was added.

4) Figure 1G; Lines 51-52: The p-value provided in the main text and that in the figure are not the same. A reference to Figure 1G is missing in line 52.

Thank you for noticing this discrepancy. The value was verified and the discrepancy was corrected.

The reference to the figure was added.

5) Figure 2B: The thick line used to mark the CIDs seems to obscure the features that may be used to visually distinguish CIDs. Could you use a thinner line? Visually, I can only confidently agree with CIDs 7 and 15 from the left.

The thick line was slimmed-down as suggested for figure 2B.

We agree that not all CIDs appear visually to the human eye. But they are statistically detected and appear strongly when looking at the changes in chromosome conformations between ActD or Cam treated samples and their controls.

6) Figure 2C: This figure looks weird: the directionality indices for the loci are overlapping. The reference for calculating directionality indices in the Methods section measures directionality preferences (is it the same?) and represents the data differently. Either the representation should be standardized, or the analysis performed for this paper should be described in more detail in the Methods section.

The overlapping was due to a representation issue that was corrected.

Thank you for noticing the discrepancy in directional index/preference nomenclature. We corrected it by opting for directional preference in this manuscript. The difference in representation from the reference and our manuscript comes from the difference in scales: because the representation in our manuscript is for the entire chromosome rather than for specific loci as in the reference, we decided to use another more appropriate representation.

7) The figures have not been referenced sequentially throughout the text. Figures S2 and S3 are not referenced in the main text. Figure S10 (line 103) is not there.

Figure numbers were updated.

8) Figure 2G: HEID and ROC terminologies are used in the figure before they are introduced in the main text.

Thank you for catching this. We have defined the terminologies in the figure legend and have directed the reader to the main text for extended description of the definitions.

9) Figure 4D-right: The triangles marking the transcription-related genes are difficult to distinguish. Marking these genes using a color that contrasts the black and grey of the figure will highlight these better.

We agree that the triangles are difficult to distinguish. However, using an additional contrasting color would remove the information about whether the RNA level of transcription-related genes is significantly changing. We tried other ways (shapes, full versus empty) for marking the transcription-related genes and could not find any that were better than the triangles.

10) Line 102-105: Here, it looks as if the authors are missing a reliable test to verify the effect of Chloramphenicol. Perhaps SDS-PAGE?

We acknowledge in the text that chloramphenicol inhibition of translation may be incomplete. However, based on analogy with the sensitivity of translation to the drug in the reasonably closely related *Sulfolobus* species and in light of the significant impairment of *Aeropyrum pernix* growth upon chloramphenicol treatment, we believe it likely that the drug is indeed targeting translation.

11) In lines 102-104 the authors say that growth is retarded in cultures treated with chloramphenicol. In lines 112-114 they suggest that chloramphenicol treatment may modestly disrupt translation. These sentences are contradictory.

Respectfully, we disagree – even a modest inhibition of translation could lead to a reduced growth rate.

12) Line 122: Perhaps the authors want to refer to Figures 5I and 5J?

Indeed. This was changed.

13) The Materials and Methods section has multiple errors. For instance, the names of the chemicals in the materials and methods are not written with the proper subscripts, the space between numbers and units is missing, for instance, 0.5M instead of 0.5 M, in some places the capital letter 'O' has been substituted by the number '0', mentions of optical density (OD) do not have an associated wavelength, on occasion 'u' has been used instead of 'μ', etc. Please go over the Materials and Methods section again and correct these.

Thank you for noticing these errors. We endeavored to correct them all.

14) Line 210-211: Why were the genomic coordinates redefined? Why were the original coordinates not used?

The redefinition of the genomic coordinates is explained in the article cited for the 3C methodology (Takemata et al., 2019): *“In our preliminary analysis of S. acidocaldarius, we noticed that interaction score was extraordinarily high between the first bin and the last bin. These bins are adjacent on the circular chromosome of S. acidocaldarius but not are neighbors based on the genomic coordinates. HiC-Pro failed to discard self-ligation junctions of the restriction fragments panning the boundary of the two bins, as it was originally developed to analyze Hi-C data from eukaryotic linear chromosomes. To solve this problem, we re-defined the genomic coordinates of each Sulfolobus species so that they started from the first base of the first restriction site in the original definition”*

The text was modified to briefly explain this.

15) The 3C-Seq experiments were done in replicate. Show the individual contact maps in the supplementary information.

A supplementary figure was added with all the individual contact maps.

16) Line 246-247: The sentence is incorrect.

The grammatical error was corrected.

17) Lines 270-271: '... Tris, pH=8. DNA. DNA libraries...' There seems to be an error here.

Corrected.

18) Line 276: '...calculated as observed-(theoretical-average) with...' Is this supposed to be a formula? It is not written so clearly.

The formula was written differently to facilitate its reading: " $n_{i,normalized}=n_{i,observed}-(n_{i,theoretical} - n_{average})$ "

19) Lines 313-314: Is the composition of TBS-TT buffer correct?

The composition of the buffer is written as used in the ChIP experiment and as published for *Sulfolobus* in Bell et al. 1999 (10.1016/S1097-2765(00)80226-9).

20) Lines 388-390: This reference has not been formatted properly.

This was corrected.

21) Split the supplementary figures into panels, and check the figure legends: Figure S8 is not split into panels as has been described in the figure legend. There is no panel J in Figure S9.

Thank you for noticing these typos.

Supplementary figures were split into panels when appropriate.

Decision Letter, first revision:

Message: Our ref: NMICROBIOL-23020341A

29th September 2023

Dear Dr. Bell,

Thank you for submitting your revised manuscript "Chromosome architecture in an archaeal species that naturally lacks Structural Maintenance of Chromosomes proteins." (NMICROBIOL-23020341A). It has now been seen by the original referees and their comments are below. The reviewers find that the paper has improved in revision, and therefore we'll be happy in principle to publish it in Nature Microbiology, pending minor revisions to satisfy the referees' final requests and to comply with our editorial and

formatting guidelines.

Thank you again for your interest in Nature Microbiology Please do not hesitate to contact me if you have any questions.

Sincerely,

Kyle

Dr. Kyle Frischkorn
(he/him/his)
Senior Editor, Nature Microbiology
Nature Portfolio

Reviewer #1 (Remarks to the Author):

The changes made significantly improve the clarity and accuracy of the results. This revised version will be of great interest to readers interested in archaea and chromosome conformation in general.

Reviewer #2 (Remarks to the Author):

Thank you for addressing my comments.

Reviewer #3 (Remarks to the Author):

I am overall very satisfied with the authors' responses to reviewer's comments. The authors received the comments well, and used them to strengthen their manuscript.

Ideally the authors could address two of the points that I raised originally, somewhat more elaborately to make their conclusions more convincing.

1) The authors approach to comment 7 in the list of specific comments I had is commendable. Plotting 3C read depth VS the position on the chromosome is indeed an easier and more straightforward way to test whether loop anchors correspond to areas of decreased read depth. I would, however, suggest addition of one more plot as supplementary figure to be entirely convincing: A plot of unique 3C reads VS position on the chromosome. As the plot is now, 3C read depth does not filter out duplicate reads that come from amplification biases during PCR of the 3C-Seq library prior to sequencing.

2) The response of the authors to comment 12 in the first revision (regarding the correlation of HEID' and ActD-resistant genes) is fine. However, I think that the slight bias for ActD-resistant genes in HEID' is not convincing enough to say that these genes actively structure HEID' (but not ROC). Therefore, I would recommend that the statistics provided in the comment be included in the main text.

Decision Letter, author guidance:

Message: Our ref: NMICROBIOL-23020341A

6th October 2023

Dear Steve,

Thank you for your patience as we've prepared the guidelines for final submission of your Nature Microbiology manuscript, "Chromosome architecture in an archaeal species that naturally lacks Structural Maintenance of Chromosomes proteins." (NMICROBIOL-23020341A). Please carefully follow the step-by-step instructions provided in the attached file, and add a response in each row of the table to indicate the changes that you have made. Please also check and comment on any additional marked-up edits we have proposed within the text. Ensuring that each point is addressed will help to ensure that your revised manuscript can be swiftly handed over to our production team.

In recognition of the time and expertise our reviewers provide to Nature Microbiology's editorial process, we would like to formally acknowledge their contribution to the external peer review of your manuscript entitled "Chromosome architecture in an archaeal species that naturally lacks Structural Maintenance of Chromosomes proteins.". For those reviewers who give their assent, we will be publishing their names alongside the published article.

Nature Microbiology offers a Transparent Peer Review option for new original research manuscripts submitted after December 1st, 2019. As part of this initiative, we encourage our authors to support increased transparency into the peer review process by agreeing to have the reviewer comments, author rebuttal letters, and editorial decision letters published as a Supplementary item. When you submit your final files please clearly state in your cover letter whether or not you would like to participate in this initiative. Please note that failure to state your preference will result in delays in accepting your manuscript

for publication.

Cover suggestions

COVER ARTWORK: We welcome submissions of artwork for consideration for our cover. For more information, please see our https://www.nature.com/documents/Nature_covers_author_guide.pdf guide for cover artwork.

Nature Microbiology has now transitioned to a unified Rights Collection system which will allow our Author Services team to quickly and easily collect the rights and permissions required to publish your work. Approximately 10 days after your paper is formally accepted, you will receive an email in providing you with a link to complete the grant of rights. If your paper is eligible for Open Access, our Author Services team will also be in touch regarding any additional information that may be required to arrange payment for your article.

Please note that *Nature Microbiology* is a Transformative Journal (TJ). Authors may publish their research with us through the traditional subscription access route or make their paper immediately open access through payment of an article-processing charge (APC). Authors will not be required to make a final decision about access to their article until it has been accepted. [Find out more about Transformative Journals](https://www.springernature.com/gp/open-research/transformative-journals)

Authors may need to take specific actions to achieve [compliance with funder and institutional open access mandates](https://www.springernature.com/gp/open-research/funding/policy-compliance-faqs). If your research is supported by a funder that requires immediate open access (e.g. according to [Plan S principles](https://www.springernature.com/gp/open-research/plan-s-compliance)) then you should select the gold OA route, and we will direct you to the compliant route where possible. For authors selecting the subscription publication route, the journal's standard licensing terms will need to be accepted, including [self-archiving policies](https://www.nature.com/nature-portfolio/editorial-policies/self-archiving-and-license-to-publish). Those licensing terms will supersede any other terms that the author or any third party may assert apply to any version of the manuscript.

Please use the following link for uploading these materials:
[redacted]

Best regards,

Kyle

Reviewer #1:

Remarks to the Author:

The changes made significantly improve the clarity and accuracy of the results. This revised version will be of great interest to readers interested in archaea and chromosome conformation in general.

Reviewer #2:

Remarks to the Author:

Thank you for addressing my comments.

Reviewer #3:

Remarks to the Author:

I am overall very satisfied with the authors' responses to reviewer's comments. The authors received the comments well, and used them to strengthen their manuscript.

Ideally the authors could address two of the points that I raised originally, somewhat more elaborately to make their conclusions more convincing.

1) The authors approach to comment 7 in the list of specific comments I had is commendable. Plotting 3C read depth VS the position on the chromosome is indeed an easier and more straightforward way to test whether loop anchors correspond to areas of decreased read depth. I would, however, suggest addition of one more plot as supplementary figure to be entirely convincing: A plot of unique 3C reads VS position on the chromosome. As the plot is now, 3C read depth does not filter out duplicate reads that come from amplification biases during PCR of the 3C-Seq library prior to sequencing.

2) The response of the authors to comment 12 in the first revision (regarding the correlation of HEID' and ActD-resistant genes) is fine. However, I think that the slight bias for ActD-resistant genes in HEID' is not convincing enough to say that these genes actively structure HEID' (but not ROC). Therefore, I would recommend that the statistics provided in the comment be included in the main text.

Author Rebuttal, first revision:

Response to Referees

We thank the referees for their constructive and helpful comments. Only referee three had requests at this stage. Our response is inserted below in red.

Reviewer #1:

Remarks to the Author:

The changes made significantly improve the clarity and accuracy of the results. This revised version will be of great interest to readers interested in archaea and chromosome conformation in general.

Reviewer #2:

Remarks to the Author:

Thank you for addressing my comments.

Reviewer #3:

Remarks to the Author:

I am overall very satisfied with the authors' responses to reviewer's comments. The authors received the comments well, and used them to strengthen their manuscript.

Ideally the authors could address two of the points that I raised originally, somewhat more elaborately to make their conclusions more convincing.

1) The authors approach to comment 7 in the list of specific comments I had is commendable. Plotting 3C read depth VS the position on the chromosome is indeed an easier and more straightforward way to test whether loop anchors correspond to areas of decreased read depth. I would, however, suggest addition of one more plot as supplementary figure to be entirely convincing: A plot of unique 3C reads VS position on the chromosome. As the plot is now, 3C read depth does not filter out duplicate reads that come from amplification biases during PCR of the 3C-Seq library prior to sequencing.

In fact, the figure that we generated for the referee was created using Hi-C Pro and duplicate reads were filtered out in our analysis.

2) The response of the authors to comment 12 in the first revision (regarding the correlation of HEID' and ActD-resistant genes) is fine. However, I think that the slight bias for ActD-resistant genes in HEID' is not convincing enough to say that these genes actively structure HEID' (but not ROC). Therefore, I would recommend that the statistics provided in the comment be included in the main text.

We have incorporated the statistics into the text (lines 138 – 140) as per the referee's request.

Final Decision Letter:**Mess** 30th October 2023**age:**

Dear Professor Bell,

I am pleased to accept your Article "Chromosome architecture in an archaeal species naturally lacking Structural Maintenance of Chromosomes proteins" for publication in Nature Microbiology. Thank you for having chosen to submit your work to us and many congratulations.

Acceptance of your manuscript is conditional on all authors' agreement with our publication policies (see <https://www.nature.com/nmicrobiol/editorial-policies>). In particular your manuscript must not be published elsewhere and there must be no announcement of the work to any media outlet until the publication date (the day on which it is uploaded onto our website).

Please note that *Nature Microbiology* is a Transformative Journal (TJ). Authors may publish their research with us through the traditional subscription access route or make their paper immediately open access through payment of an article-processing charge (APC). Authors will not be required to make a final decision about access to their article until it has been accepted. [Find out more about Transformative Journals](https://www.springernature.com/gp/open-research/transformative-journals)

Authors may need to take specific actions to achieve [compliance](https://www.springernature.com/gp/open-research/funding/policy-compliance-faqs) with funder and institutional open access mandates. If your research is supported by a funder that requires immediate open access (e.g. according to [Plan S principles](https://www.springernature.com/gp/open-research/plan-s-compliance)) then you should select the gold OA route, and we will direct you to the compliant route where possible. For authors selecting the subscription

publication route, the journal's standard licensing terms will need to be accepted, including [self-archiving policies](https://www.nature.com/nature-portfolio/editorial-policies/self-archiving-and-license-to-publish). Those licensing terms will supersede any other terms that the author or any third party may assert apply to any version of the manuscript.

With kind regards,

Kyle

Dr. Kyle Frischkorn
(he/him/his)
Senior Editor, Nature Microbiology
Nature Portfolio

P.S. Click on the following link if you would like to recommend Nature Microbiology to your librarian <http://www.nature.com/subscriptions/recommend.html#forms>

** Visit the Springer Nature Editorial and Publishing website at http://editorial-jobs.springernature.com?utm_source=ejp_NMicro_email&utm_medium=ejp_NMicro_email&utm_campaign=ejp_NMicro for more information about our career opportunities. If you have any questions please click [here](mailto:editorial.publishing.jobs@springernature.com).**